# ShapeY: A Principled Framework for Measuring Shape Recognition Capacity via Nearest-Neighbor Matching

## Abstract

In humans, the ability to recognize and categorize objects despite variations in 3D viewpoint, lighting, surface properties, optical conditions, etc., is known to depend primarily on shape. In contrast, state-of-the-art artificial vision systems based on deep networks rely on an intricate mixture of shape, texture and color cues that defies simple characterization. To better understand the capabilities of machine vision systems specifically with regard to shape, we developed ShapeY, a novel recognition benchmark that uses nearest-neighbor matching of images containing only shape information to probe the structure of an OR system's embedding space from a shape recognition perspective. ShapeY consists of 68,200 grayscale images of 200 3D objects rendered from multiple viewpoints, and optionally subjected to distracting non-shape "appearance" changes (color, texture, lighting, etc.). Quantitative and qualitative performance readouts include error rates for category vs. object-level matching, viewpoint tuning curves, histograms of matching scores to positive vs. negative exemplars, and grids showing ordered best matches. We computed ShapeY's metrics for 321 pre-trained networks with diverse architectures, and found substantial variation in performance depending on model type, size and training scheme. Interestingly, even the best performing system (a ViT trained on DINO V2), which displayed excellent matching behavior "on average", produced egregious matching errors on occasion, presumably reflecting "tangling" of its embedding space at the fine scale. By providing a detailed profile of an OR system's shape representing capabilities, ShapeY can help track progress of machine vision systems along the dimension that aligns most closely with human recognition behavior.

## 1 Introduction

The fundamental problem of object recognition (OR) in the natural world can be described simply: after seeing an object or scene under one set of viewing conditions, the OR system should be able to recognize the same object or scene later under different viewing conditions, such as from a different distance, viewing angle, lighting conditions, etc. This basic visual ability allows an animal to successfully store and retrieve food, to recognize its home despite approaching it from different directions, and to escape quickly to known safe locations. Put another way, to a competent OR system, two different views of an object or scene should "look alike" (i.e. map to nearby points in the system's embedding space; this is our core premise), and conversely, two images that look alike, that is, are near to each other in the OR system's embedding space, should be images of the same object or scene (or nearly so) (Figure 1).

In humans, such similarity judgements are known to depend primarily on shape, more than other perceptual features such as color, texture, or size (Grill-Spector et al., 2001; Biederman, 1987; Biederman & Ju, 1988; Hoffman, 1998; Kourtzi, 2001). Critically, shape cues, or contour features that remain stable across changes in viewpoint, appear to be closely tied to this capacity for one-shot similarity judgement, permitting novel objects to be judged similar without prior exposure (Biederman & Bar, 1999). Consistent with this, shape-based recognition comports with the fact that "basic level" categories (e.g. tree, bird, chair), which lie at the

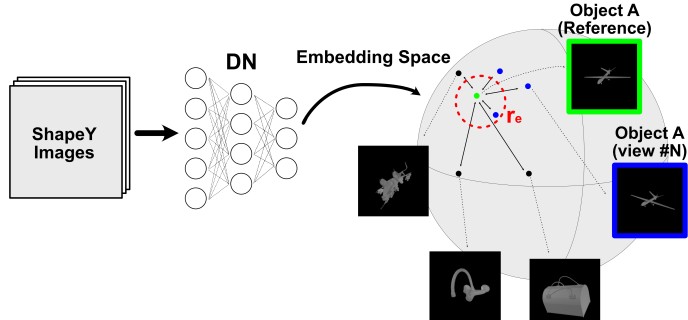

**Task: Choose an image that looks most similar to the reference image.**

Figure 1: Overview of how ShapeY tests shape understanding in object recognition systems. Our core premise is that any image of an object in slightly transformed viewpoints (blue dots) should land in proximity to the reference view (green) in the embedding space, closer than images of all other objects (black). ShapeY employs nearest-neighbor matching in the embedding space while excluding some of the views physically closest to the reference (red circle) to assess the shape understanding capability.

heart of human cognition, group together objects that are roughly similar in shape.[1] Given the strong shape bias in human visual perception, and the central role shape understanding plays in supporting generalizable object recognition, it is surprising that the benchmark task most often used to rate the performance of artificial OR systems—ImageNet (Deng et al., 2009)—diverges from a shape-based same-object recognition task in three important ways.

First, images in many ImageNet categories, considered as wholes, i.e. without segmentation, have little in common in terms of shape. Even categories with names suggestive of very specific object classes, e.g. "beer bottle", would be more appropriately labeled as, "Scenes containing one or more beer bottles and other objects". Further, even when the depicted objects are considered in isolation, the class instances can vary in shape from image to image, in uncontrolled amounts, especially for non-rigid objects such as animals. This leads to questions such as, "Should two snakes in different body configurations, and thus having different shapes, be considered the same or different objects?" (After all, if that much shape variation is tolerated for snakes, should it not also be tolerated for cars and spoons?). Given these complications, Imagenet is best described as a "superordinate" classification task, in which the images that must be classified together as wholes have a degree of abstract similarity, but share little in common in terms of *overall* shape, and even the isolated category instances often vary significantly in shape.

Second, even in classes when the class-named objects are relatively consistent in shape (e.g. mostly true in the "beer bottle" class), the classification task combines two different types of challenges, again in uncontrolled amounts: (1) segmenting the class-named objects from the overall scene, which varies in difficulty from scene to scene depending on the number and arrangement of class-named and other objects, and (2) mapping the segmented objects to nearby points in the OR system's embedding space, that is, "recognizing" them. This mixing of tasks makes it difficult to attribute classification success or failure to the relative efficacy of the OR system's attentional vs. representational mechanisms.

Third, ImageNet images are replete with non-shape cues, including color, texture and context, making it difficult to know the degree to which an OR system's classification performance depends on intra-class similarity of shape vs. other types of cues. While relying on non-shape cues may be necessary in certain contexts (e.g., recognizing a snake in the wild regardless of its body configuration), an OR system that

---

[1]Besides being tightly bound to shape, basic level categories are special in other ways: they are the first categories learned by children; they are the most frequently used in human language; they are mostly assigned short words; and they allow for the fastest access to mental representations (e.g., "bird" is more rapidly processed than "sparrow").

depends too heavily on non-shape cues is at risk of being fooled: a shoe made of snake skin should not be classified as a snake, for example.[2]

In light of these complications, and in recognition of the importance of shape in human object recognition, a handful of novel training and/or testsets have been developed that more directly explore the degree to which shape influences an OR system's ImageNet classification performance. For instance, Baker et al. (2018) demonstrated that classification accuracies of deep network (DN)-based OR systems are significantly affected when objects are presented as silhouettes or in unusual textures or styles. Geirhos et al. (2019) expanded on this idea by creating "cue-conflict" images, where texture and contour provide conflicting signals for object class determination. Their findings revealed that many DNs exhibit a texture bias, in contrast to humans who rely primarily on shape. This difference between human and machine vision has led to numerous efforts to induce a shape bias in DNs in the hopes of improving their robustness (Müller et al., 2024; Gavrikov et al., 2024; Li et al.; Hosseini et al., 2018; Islam et al., 2021; Jarvers & Neumann, 2024; Hemmat et al.; Mummadi et al., 2021; Gavrikov & Keuper). However, this literature operationalizes shape sensitivity almost entirely in terms of robustness to texture or cue-conflict manipulations within a single, static image, rather than asking whether an OR system's shape-based representations remain stable across the changes in viewpoint that define the object recognition problem itself. It is this latter property, view invariance, that our work is designed to assess.

In summary, while the ImageNet task is challenging, and has helped to drive the development of highly capable DN-based OR systems, the ImageNet classification task differs in multiple ways from the problem that biological vision systems evolved to solve (i.e. matching views of the same object or scene under different viewing conditions), and its performance measures provide little direct information about an OR system's capacity for matching based on shape, the dominant cue used in human vision.

What should a database of images look like, and what testing approach should be used, to best assess an OR system's ability to use shape information to solve the fundamental problem of OR? In an effort to address these questions, our main contributions in this paper are as follows:

- We introduce ShapeY, an integrated dataset and benchmarking system that assesses the quality of an OR system's embedding space for shape-based recognition using nearest neighbor matching of object views.

- We provide a suite of quantitative and graphical tools for analyzing the fine-grained structure of an OR system's embedding space.

- We use ShapeY to analyze a variety of deep networks, offering insights into the characteristics of systems with stronger shape understanding capabilities.

## 2 The ShapeY Image Set and Matching Task

### 2.1 Desirable properties of the image database

Regarding the ideal image database to test 3D shape-based recognition capability, we considered seven properties to be important:

1. The images should depict views of the same or similarly-shaped 3D objects shot under different viewing conditions (pose, lighting, etc.).

2. The objects should be rigid to avoid the same-different shape ambiguity that arises when shape deformations are allowed.

---

[2]Various authors have attempted to quantify the relative contributions of shape vs. other cues to classification performance, and have concluded that conventional deep network (DN)-based OR systems tend to be texture, rather than shape-biased (Geirhos et al., 2019; Brendel & Bethge, 2019; Baker et al., 2018).

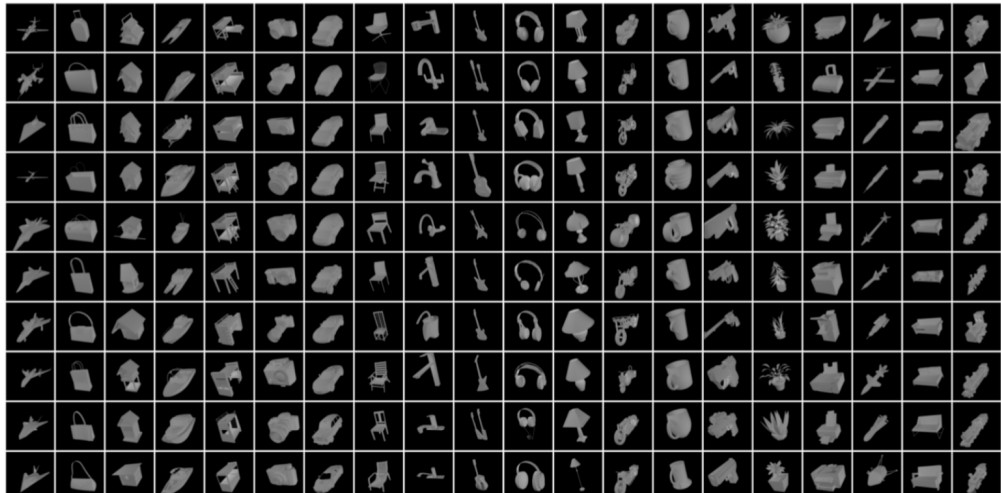

Figure 2: Complete set of 3D object models rendered using Blender at their "origin viewpoints". Objects are grouped into 20 categories with 10 instances of each.

3. The objects should be isolated (i.e. not occluded, camouflaged, or embedded in cluttered scenes) in order that recognition performance be a pure reflection of shape representation capability.

4. The images should be shot under favorable imaging and lighting conditions to ensure that recognition performance reflects a system's representational capabilities, rather than its capabilities at the pixel-processing-level.[3]

5. All non-shape information, including color, texture and lighting of both object and background surfaces should be manipulable, both to thwart the use of non-shape cues in the recognition process, or to allow such cues to be used as distractors.

6. The images should be organized into basic level categories, making it possible to assess an OR system's ability to distinguish objects with varying degrees of shape inter-similarity, e.g., distinguishing airplanes from cars (a basic level distinction), distinguishing different types of airplanes from each other (a subordinate level distinction), and distinguishing – or matching – different views of the same object.

7. The set of objects and views contained in the database should be large enough so that there is a reasonable chance of detecting failure cases where nearby points in the embedding space correspond to obviously completely different (OCD) objects, if such failures exist.

It is worth noting that properties 2-4 represent simplifications relative to the natural vision problem, which could be a basis for criticism of an OR benchmarking system. However, as we show below, the problem of equating 3-D object views across changes in viewpoint and other viewing conditions remains challenging to state-of-the-art OR systems, even under these simplified conditions.

Several image databases that have some, but not all of these features have been created to serve as benchmarks for OR systems (Table 1). iLab20M has all but one of these features: 20 million images of 718 basic level rigid object classes, all toy vehicles, shot at systematically varying 3D viewpoints under different lighting conditions, optical conditions, and on different backgrounds. However, object colors and other surface properties are not manipulable, given that iLab20M images are photographs of actual toy vehicles. In

---

[3]Shape information in the ShapeY image set mainly resides in the object contour structure (i.e. information contained in the line drawing): contour position, orientation, curvature, and junctions (sharp angle changes, and curvature changes). For simplicity, the shape information in ShapeY's images should be clean and clear: object contours are not degraded by, e.g., a contrast attack, low light, low contrast, fog, etc., and we include no shape noise such adding object clutter to the images, or viewing the object through an intervening hedge, solid occlusion, or painting a camouflage pattern onto the object surface.

Table 1: 3D view-based image datasets. We list a subset of desired properties in the table.

| Dataset | #Objects | #images | Categorical hierarchy | Non-shape cues | Viewpoints |
|---|---|---|---|---|---|
| COIL (Nene et al.) | 100 | 7k | Yes | Uncontrolled | Controlled |
| NORB (LeCun et al., 2004) | 50 | 194k | Yes | Controlled | Controlled |
| RGB-D (Lai et al., 2011) | 300 | 250k | Yes | Uncontrolled | Controlled |
| Big-BIRD (Singh et al., 2014) | 125 | 75k | Yes | Uncontrolled | Controlled |
| iLab-20M (Borji et al., 2016) | 718 | 22M | Yes | Uncontrolled | Controlled |
| CORe50 (Lomonaco & Maltoni, 2017) | 50 | 165k | Yes | Uncontrolled | Uncontrolled |
| Objectron (Ahmadyan et al., 2020) | 14k | 3.9M | Yes | Uncontrolled | Uncontrolled |
| CO3D (Reizenstein et al., 2021) | 19k | 1.5M | Yes | Uncontrolled | Uncontrolled |
| PUG (Bordes et al., 2023) | 724 | 88k | Yes | Varied | Controlled |
| ShapeY | 200 | 68k | Yes | Controlled | Controlled |

ShapeY, we elected to use graphically rendered images from 3D models as described below, both to provide a pathway for efficient expansion of the database (given that an increasing number of 3D object and scene models are becoming freely available), and to provide maximum flexibility for the manipulation, in software, of lighting and surface cues as the need arises.

It is important to add that our goal in this work has been to create a database for use in evaluating a pre-trained (or pre-designed) OR network's ability to recognize views of novel 3D objects despite changes in viewing conditions, similar to recent studies on robustness of OR systems (Hendrycks & Dietterich, 2019; Wang et al., 2019; Bordes et al., 2023). This kind of "one shot" recognition capability is a clear feature of human vision (Biederman & Bar, 1999; Biederman, 1981; Biederman & Ju, 1988), and a pre-requisite for functioning reliably in real-world situations. As such, the ShapeY image database is not intended to be used as a training set, though in an exception to this, we carry out a few linear probing and fine-tuning experiments using the image database (see Section 3.8), both as a way of ruling out the "out of distribution" (OOD) hypothesis, and to better understand the performance characteristics of a representative DN-based OR system.

Having laid out several desirable properties of an image database, we now turn to the way we will use the image set to assess a vision system's capacity for shape-based OR. Our testing approach is closely tied to the fundamental problem of OR, requiring that (1) views of the same object seen under different viewing conditions reliably map to nearby points in an OR system's embedding space, that is, different views of the same object should "look alike" to the system, and (2) nearby points in the embedding space, that is, inputs that "look alike" to the system, should reliably correspond to images of the same or similarly-shaped objects. Or, at a minimum, nearby embedding vectors should never be associated with "obviously completely different" objects. Simultaneous satisfaction of these two requirements indicates that an OR system has a well-formed embedding space for shape-based recognition.

## 2.2 Constructing the ShapeY image set

The ShapeY image set contains a total of 68,200 views of 200 objects. The images are produced using Blender and publicly available 3D models from the ShapeNet collection (Chang et al., 2015). In the default database, each 256x256 image depicts a single object, texture-less and gray in color, shown against a black background (Figure 2).

The database is structured as a 3-level hierarchy. The top level of the hierarchy consists of 20 basic-level categories (airplane, bunkbed, faucet, plant, etc). Each category contains 10 objects, and each object is represented by a set of 341 views covering five degrees of freedom of rigid viewpoint transformation (**p**itch, ya**w**, **r**oll, **x**, **y**), centered on an "origin" view. The origin view of each object shown in Figure 2 is standardized in size to cover a maximum of roughly two-thirds of the image's width or height, and standardized in orientation to show as much shape detail as possible. Views of each object are grouped into viewpoint "series" (Figure 3). Each series consisting of 11 views is generated by incrementally applying a

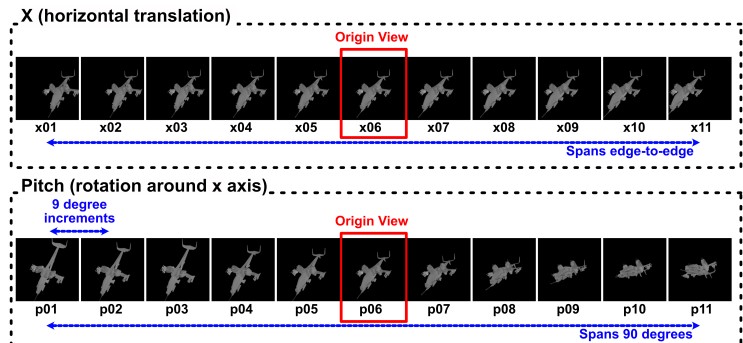

Figure 3: Examples of how an object is transformed to construct different series. For translation (top), the object is shifted from one edge of the image to another. For rotation along an axis (bottom), the object is rotated in 9° increments, spanning 90° total across 11 images.

specific viewpoint transformation (VT) to the origin view, in five steps moving away from the origin view in both directions. The 3 rotational transformations ($\mathbf{p}$, $\mathbf{r}$, $\mathbf{w}$) are individually limited to a maximum of 90 degrees of rotation over the 11-image series (in steps of 9°) to avoid scenarios where the visible 3D structure of an object changes drastically from its origin view. Translation steps are roughly 3.3% of the frame width so that the object is almost always entirely visible in the frame across each series. Scale changes are excluded in the current version of the database to preserve object detail that would be lost at smaller scales. Some series include combinations of VTs, to create greater viewpoint diversity. For example, in the series combining "x" and "roll" ($\mathbf{xr}$), each step in the series advances by both a rightward horizontal shift and a clockwise image plane rotation. In series containing more than one rotation transformation, we rotate in the order indicated by the series name (rotation operations are not commutative). For example, in the **prw** series, we rotate in pitch first, followed by roll and then yaw. Examples of all series containing **p** and **w** are shown in Figure 4. Combining viewpoint transformations leads to 31 possible VTs ($31 = 5$ transformations chosen 1, 2, 3, 4, or 5 at a time) with 11 images each, leading to 341 views per object.

In addition to showing invariance to viewpoint changes, a shape-based OR system must also be able to ignore other types of changes in an object's projected image that do not involve an underlying shape change. This includes changes in the color, texture, and specularity of an object's surfaces, changes in the intensity, color and pattern of lighting, and changes in image contrast and sharpness due to optical/atmospheric effects (smoke, fog, etc.). We refer collectively to all non-shape (non-viewpoint) changes as "appearance" changes. The way these appearance-altered images are used to measure recognition performance will be discussed below.

### 2.3 The nearest-neighbor-based ShapeY matching task

Given a reference image, the ShapeY matching task asks an OR system to rank order all images in the database based on their similarity score to the reference image in the OR system's embedding space using cosine similarity or another similar measure. A response is scored as "correct" if the closest match is to another (eligible) view of the same object; "categorically correct" if the closest match is to an eligible view of a different object within the same category; and "incorrect" if the closest match is to a distractor from a different object category (Figure 5).

What determines eligibility for positive matching? Because we render images densely in viewpoint space, so that many images differ from each other by only a small change in viewpoint, a view that differs only slightly from the reference view would essentially always match the reference view best, leading to a perfect nearest-neighbor classification score in every case (which would provide zero information). Therefore, instead of including every view of a reference object in its set of positive match candidates (PMCs), we select which views to include as candidates following the procedure described next.

Positive match candidates

Figure 4: Positive match candidates (PMCs) for view #8 (blue box) out of 11 in the series with CVT = 'pw'. Rows show all 8 series containing 'pw'. The difficulty of the matching task is controlled by excluding positive match candidates in the "vicinity" of the reference view in viewpoint space. The "exclusion zone" shown (red shading) is for an exclusion radius $r_e = 2$.

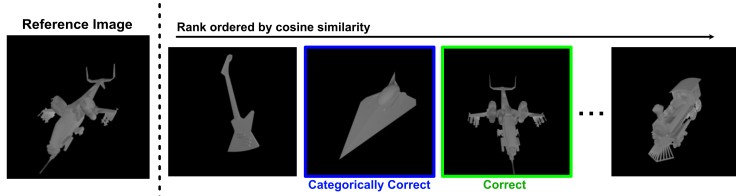

Figure 5: Example of a matching task. Given a reference image, we rank order other images based on their cosine similarities in the embedding space of an object recognition system. If the best matching image is a view of the same object (green), it is considered correctly matched; if the best matching image is a view of an object in the same category (blue), it is considered a categorically correct match; and if the best matching image is a different object, it is considered an incorrect match.

First, we break down performance scoring by VT ($\mathbf{xy}$, $\mathbf{p}$, $\mathbf{w}$, $\mathbf{pw}$, etc.), since an OR system may have uneven competence in coping with different types or combinations of viewpoint transformations. Having chosen a particular VT, say $\mathbf{pw}$, the set of reference images to be scored includes only the $200 \times 11 = 2{,}200$ object views contained in $\mathbf{pw}$ series. We also limit the set of PMCs for each reference image to include only images of the reference object in series whose VTs include all transformation axes of the chosen VT. For the $\mathbf{pw}$ test, for example, the PMCs would come only from the series $\mathbf{pw}$, $\mathbf{xpw}$, $\mathbf{xypw}$, $\mathbf{xprw}$, $\mathbf{xyprw}$, $\mathbf{ypw}$, $\mathbf{prw}$, and $\mathbf{yprw}$.

Second, we exclude all object views from the PMC that are too close to the reference view, that is, that fall within a defined viewpoint "exclusion radius" ($r_e = 0, 1, 2, ...$). Note that "no exclusion" and $r_e = 0$ are distinct conditions: under no exclusion, every view of the object other than the identical reference image itself is an eligible positive match candidate, whereas $r_e = 0$ already begins the exclusion procedure by removing, across all eligible series, the view at the reference's own series index—that is, the most viewpoint-similar

poses, which include the reference image itself. This is the reason we index the exclusion starting at $r_e = 0$: it denotes the exclusion of images that fall within a radius of 0 in our constructed series, a set that already includes the reference image. For example, if the reference view is the 6th view in the pw series, and $r_e = 2$, we would exclude all PMCs from the eligible series (**pw**, **xpw**, **prw**, **xprw**, **xyprw**, **xypw**, **yprw**, **ypw**) whose indices are 4, 5, 6, 7, and 8, leaving as eligible the views 1, 2, 3, 9, 10, and 11. These restrictions to the set of PMCs in this case (**pw**, $r_e = 2$) guarantee that any successful match to a reference view must have bridged at least a $3x9 = 27°$ change in both pitch and yaw, and may also differ from the reference view by 3 or more viewpoint steps along one or more other viewpoint dimensions (in this case, **x**, **y**, and **r**). A complete set of PMCs for $r_e = 2$ in **pw** is displayed in Figure 4. This allow us to break down the matching performance score by VT per each exclusion radius, enabling an examination of how sensitive the embedding space is to specific amounts of transformation along the VT.

Note that when tallying category-level classification scores (i.e., counting how often a reference view is best-matched to any object within the same category), we apply the exclusion radius to the views of all 10 objects within the same category. Thus, for the VT condition **pw** with $r_e = 2$, the exclusion guarantees that any successful match to a reference view of say, a chair, must have bridged at least a $27°$ change in both pitch and yaw to successfully match any chair. Thus, while category-level matching is easier in the sense that the set of PMCs is 10 times larger than in object-level matching, the same viewpoint exclusion applies to all PMCs in order that the category error also accurately reflect the OR system's ability to match across viewpoint changes. This prevents the cheat that a similar looking chair in the same pose as the reference view scores the best match, leading to a correct answer without the OR system having to cope with a viewpoint change.

In addition to administering view-exclusion matching tasks, we utilize the appearance-altered images to force OR systems to match across an appearance change. For example, we create "contrast exclusion" tasks (the only appearance exclusion currently implemented), in which object views rendered in the original format with black backgrounds can only be matched to views of the same object rendered on light backgrounds. That is, views rendered on the original black backgrounds are excluded from the set of positive match candidates.

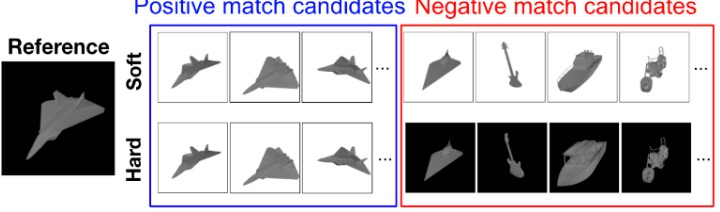

Figure 6: Examples of soft and hard contrast exclusions. In both soft and hard cases, positive match candidates are only available with the contrast-reversed background. In the soft case (top), all negative match candidates are also displayed with contrast-reversed backgrounds, while in the hard case (bottom), negative match candidates are shown with the original background, making them more "attractive".

In practice, we create two, sligthly different, contrast exclusion tasks. In the "hard" version of the contrast exclusion task, given a reference view, all same-object views with dark backgrounds are excluded as positive match candidates, forcing the system to recognize the same shape despite the change in background. All views of *other* objects are not subject to the exclusion, however, and in fact are available to match in the original black background only. In the "soft" (easier) version of the task, *all* match candidates are subject to the exclusion, and must be matched with a modified background (Figure 6).

## 2.4 Geometric perspective of ShapeY performance measures

When a view of an object or scene changes in any way, this leads to an offset in an OR system's embedding space. When the size of this offset is used as a measure of the perceptual similarity of the two views, a crucial question becomes which types of image changes lead to what sizes of offsets.

In the conceptual limit where an embedding space is populated with every view of every possible 3D object, or more realistically, a dense sampling thereof (which the ShapeY database is intended to be), a nearest

neighbor matching approach provides a means of comparing the sizes of offsets caused by non-shape changes (viewpoint and appearance), which one hopes will be small, to the sizes of offsets caused by bona fide changes in object shape, which one hopes will be large. For example, consider a reference view of a desk chair in a 3/4 pose. If the offset in embedding space caused by 30 degrees of depth rotation is greater than the offset caused by changing to a view of a rocking chair in a 3/4 pose, a nearest neighbor matching error will occur, signaling that a practical limit of the OR system's viewpoint invariance has been reached. Measuring the probability of such errors averaged over all possible reference views, and broken down by degree and type of viewpoint perturbation, provides a rich quantitative portrait of the quality of an OR system's embedding space.

A more fine-grained geometric view provides further insight. In the vicinity of a reference view (represented by a point P in the embedding space), the set of PMCs for that reference view and a particular VT and exclusion radius (e.g. **pw**, $r_e = 2$) can be viewed as an irregularly shaped "cloud" of points surrounding $P^4$. The closest of these PMCs to P establishes the radius of the largest full-dimensional ball centered on the reference view whose contained points all lie closer to the reference view than any positive match candidate. The quality of the embedding space in the vicinity of P can then be gauged in terms of the "purity" of the contents of that VT ball.

Specifically, if the ball contains even one view of even one other object seen from any viewpoint, the nearest neighbor match for that reference view will fail, leading to an object-level error. If the ball contains views of other objects, but only from the same category as the reference view, the object-level match fails but the category-level match succeeds. If the ball contains one or more exemplars from other categories, a category-level error occurs. The overall performance of the OR system can thus be quantified in broad terms by tabulating the fraction of reference images whose VT balls contain object and/or category-level "impurities", as a function of the chosen VT, exclusion radius, and appearance exclusions, if any (Figure 7). The number of such impurities is also important—the more the worse, and these can also be quantified (Figure 8).

## 2.5 Qualitative assessment of ShapeY matching errors

While ShapeY's quantitative matching scores reflect the quality of the embedding space in aggregate, visually inspecting an OR system's matching errors provides additional insights into the fine-scale structure of the system's embedding space. These errors, which can be culled from ShapeY's pictorial nearest match grids, are of four main types, ranging from those that are inevitable but inconsequential, to those that reflect a severe "entanglement" of the OR system's embedding space (DiCarlo & Cox, 2007).

1. **Unreasonably similar objects.** If two or more objects are highly similar in shape, then all views of such objects will suffer from a high matching error rate. This occurs because the viewpoint exclusion, which makes matching more difficult, only applies to views of the same object for object-level scoring, and members of the same category for category-level scoring. As such, the best match to a reference view will always be to a nearly identical view of a practically indistinguishable object, and yet will be scored as an incorrect best match.

2. **Degenerate viewpoint errors.** Matching errors can occur when two objects of different shape become indistinguishable due to foreshortening or self-occlusion, such as when the top views of a desk and a table both degenerate to simple rectangles. Errors of this type are inevitable, but relatively rare.

3. **Shape resolution limit errors.** When objects of similar shape are confused, such errors are useful indicators of the limits of the OR system's shape discrimination power. The two levels of ShapeY's performance scoring (categories and objects) both measure VT ball purity, but at different levels of resolution. The category level scores a VT ball as pure as long as it only contains instances from a single basic level category—the loosest possible grouping of object views that still have

---

[4]This "cloud" consists of numerous 1-D "strings" of views radiating outward from P, each corresponding to one of the two ends of each eligible series, on either side of the excluded range (Figure 1)

a commonality of shape. The object level scores a VT ball as pure as long as it only contains views of a single object—the tightest possible grouping of object views from a shape perspective, all corresponding to the identical shape. Shape resolution limit errors make up the bulk of errors for a typical OR system.

4. **OCD errors.** OCD errors are cases where the best match to a reference view is a view of an object whose shape is "obviously completely different" from that of the reference object (Figure 19). This type of impurity in a VT ball points to a radical mis-ordering of shape similarity, and violates the requirement that views of objects that "look alike" to an OR system, that is, that map to nearby points in embedding space, should correspond to objects with the same or very similar shape.

To the extent that unreasonably similar objects, or accidentally similar views, are found in the database, overall error rates will be artificially inflated. However, such errors currently account only a small fraction of the error rates reported below. More importantly, these cases are "forgivable", and do not signal a breakdown in the quality of the embedding space. On the other hand, the latter two types point to representational limitations of the OR system. OCD errors in particular should never occur in a well-formed embedding space, and signal that the OR system's embedding space is extremely tangled at the fine scale. ShapeY, in its suite of tools, create a display that collects these matching errors, allowing closer visual inspection (see Figure 12).

## 3 Results

This section is organized as follows. First, we demonstrate the use of ShapeY's tools by analyzing a ResNet50 pre-trained on ImageNet1k. We then investigate the causes of ResNet50's poor performance on ShapeY through linear probing and fine-tuning experiments. Finally, we measure the performance of 321 pre-trained DNs on the ShapeY benchmark with the goal to understand which DN architectures and training approaches lead to the best shape recognition performance.

### 3.1 Depth rotations significantly reduces the matching performance of a pre-trained ResNet50

We first test the performance of a ResNet50 (He et al., 2015), pre-trained on ImageNet, which we downloaded from *timm* python library (recipe a1) (Wightman, 2019). Results of a basic matching test are shown in Figure 7. Error rates averaged over all 200 objects are shown in Figure 7 for all exclusion transformations involving either 1, 2, or 3 transformation dimensions (columns 1-3, respectively).

We observe that ResNet50 suffers more when rotations in depth are involved (**p** and **w**), while being mildly affected by image plane rotation (**r**) and minimally affected by translational axes (**x** and **y**). For the single transformation dimension **p**, the error rate is already 35% for $r_e = 2$, corresponding to an enforced 27° change in object pitch. When pitch and yaw were combined (**pw**), the error rate climbed to nearly 45% at $r_e = 2$. The near perfect invariance for **x** and **y** at $r_e = 2$ is expected given the ResNet50 embedding is taken from the global average pooling layer, which explicitly pools across image shifts. Note that because we translate the object in three-dimensional axes, the object slightly rotates from the camera's perspective. We speculate that this causes the increase in the error rate for $r_e > 4$ in **x** and **y**.

While the error rates effectively summarize the ResNet50's tolerance to viewing angle changes, they do not reveal whether the matching errors are due to a single or multiple objects in the negative match candidate set scoring higher than the top positive match. Figure 8 provides a more detailed measure of the network's performance by showing the ranks of the top positive match candidates relative to other match candidates. We observe that the top positive match ranks higher than 10 more than 20% of the time at $r_e = 3$ in **pr**. Given that there are only 10 exemplars in each object category, this indicates that the required transformation perturbs the embedding enough to cause more than 20% of cases to match images from different categories as similar. This shows that the high matching error exhibited by ResNet50 is due to its severely entangled embedding space, rather than a single view accidentally landing closer.

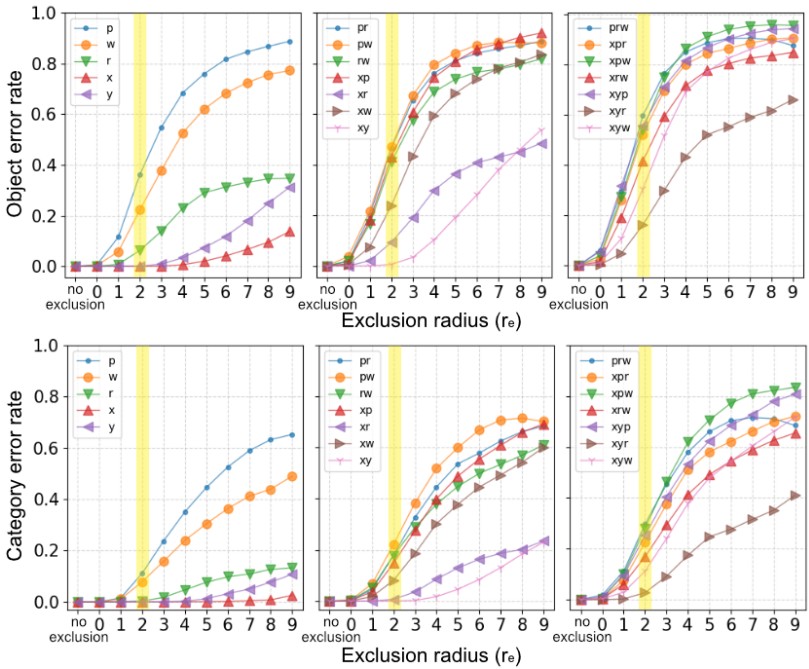

Figure 7: Nearest neighbor matching error against the exclusion radius $r_e$, for a pre-trained ResNet50. "No exclusion" denotes the condition where every non-identical view of the object is an eligible positive match candidate, distinct from $r_e = 0$, which already excludes the most viewpoint-similar poses (see main text). Top row shows errors for object matching; bottom row shows errors for category matching. Three columns show results for exclusion transformation sets with 1, 2, and 3 viewpoint transformation dimensions, respectively.

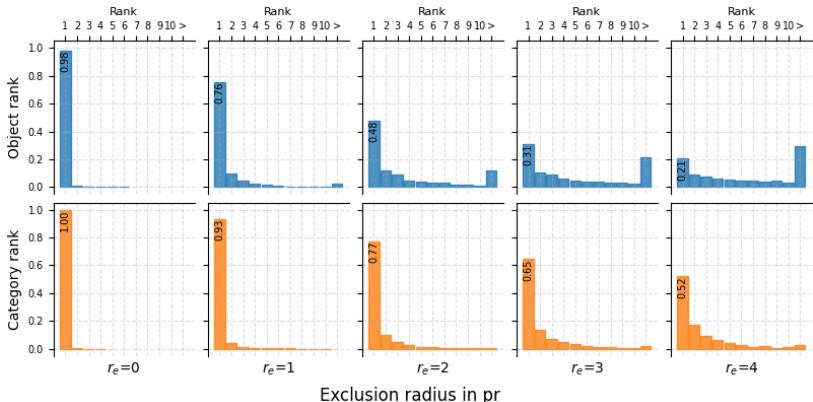

Figure 8: Rank histograms showing the distribution of ranks for the top positive match candidates, for a pre-trained ResNet50. Candidates are ranked against all other images. The top row displays results for object matching, where ranks are determined by the number of objects with higher similarity scores than the top positive match. The bottom row shows results for category matching, with ranks based on the number of categories scoring higher than the top positive match.

### 3.2 The embedding space of the ResNet50 is compartmentalized roughly into object categories

Category error rates are lower but remain substantial. For example, a 27° change in **pr** leads to nearly a 20% category error rate, meaning that 1 in every 5 views in the database is judged to be most similar to a view from an entirely different object category. It is important to rule out, however, that the better performance

in the category (compared to the object) matching task is simply due to the 10-fold increase in the number of positive match candidates available to match each input image, since matches could be made, and scored as correct, to all 10 objects within the category.

As a control, we repeat the category-matching experiment with randomized categories, each grouping 10 objects drawn from different original categories. This maintains the same number of positive match candidates as in the original test but destroys the similarity structure within each category. The category matching error rates under this manipulation are much higher than with structured categories and only slightly lower than those seen in the object matching experiments. This indicates that (1) adding a large number of unrelated positive match candidates, scattered throughout the embedding space, provides almost no benefit in our nearest neighbor matching task, and (2) ResNet50's better matching performance in the category task arises, as hoped and assumed, from the clustering in its embedding space of object views from the same object category.

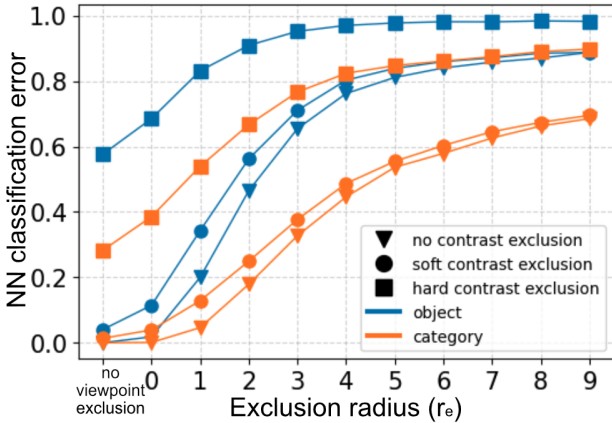

Figure 9: Nearest-neighbor matching errors when a "contrast exclusion" is compounded with viewpoint exclusions, for a pre-trained ResNet50. Object and category error rates for no, soft, and hard contrast exclusions. Results shown is for the exclusion transformation set 'pr'. The first point on the x-axis means that *all* contrast-reversed views, including the reference view itself, are available as positive match candidates.

### 3.3 A simple background change further degrades matching performance

We next test the matching performance of ResNet50 with the added challenge of a "contrast exclusion". Figure 9 shows results for the exclusion transformation **pr** using both object and category matching criteria. When a reference view can only be matched to contrast-reversed views (soft version), the category error rate increases from 19% to 25% at $r_e = 2$, and the object error rate increases above 50%. The difficulty encountered by ResNet50 in matching contrast-reversed views is striking: even when the positive match candidates for a reference view include *the exact same object view* but for the change in background, the reference view is falsely matched nearly 60% of the time to another object and 30% of the time to an object from an entirely different category.

### 3.4 The separation between the positive and the negative match candidates in the embedding space governs the matching performance

How sensitive is ResNet50's performance to enforced 3D viewpoint transformations? We plot the tuning curve to illustrate the cosine similarity decay along the transformation axis **pr** in Figure 10. Generally, the similarity score decreases with larger transformations. However, understanding how this sensitivity impacts the network's performance in our matching task requires comparing the declining similarity scores caused by viewpoint changes to the scores of the best matching negative match candidates.

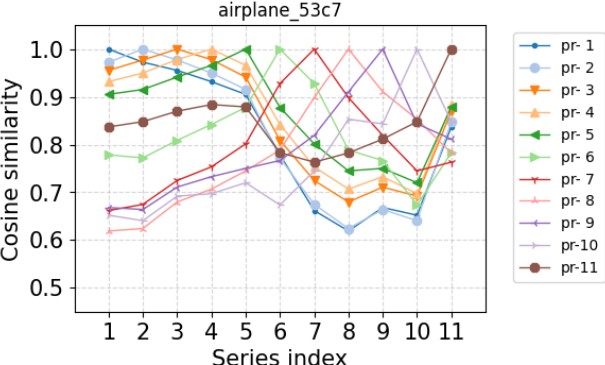

Figure 10: Tuning curves showing how each of the image in the **pr** series of an airplane is correlated to all other images in the series, for a pre-trained ResNet50. The legend reads the reference image used for each of the curve. Generally, the similarity score drops with an increment of transformation applied.

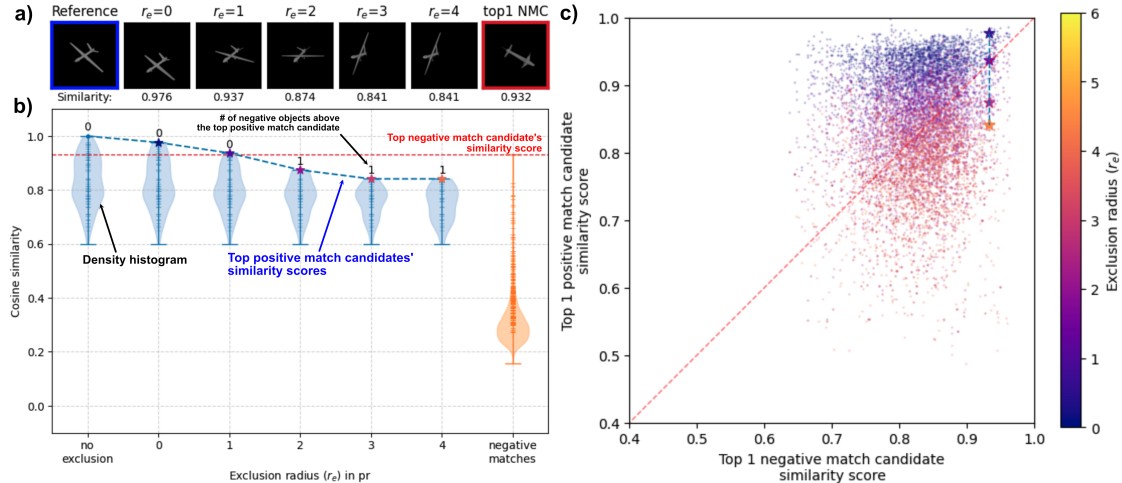

Figure 11: Comparison of cosine similarity scores for top positive and negative match candidates, for a pre-trained ResNet50. **(a)** Reference image, its best matching positive candidates for each exclusion radius in **pr**, and the top negative match candidate. **(b)** Histograms of cosine similarity scores for all positive and negative match candidates per exclusion radius in **pr** for the reference image in (a). Blue and orange lines mark scores for positive and negative candidates, respectively. The blue dotted line connects the top positive match scores per $r_e$. The red dotted line marks the top negative match score. A match error occurs when the blue line crosses the red line. **(c)** Scatter plot of top negative vs. top positive match scores for all reference images in the **pr** series, colored by exclusion radius. The red dotted line marks $y = x$; points below this line indicate match failures. Stars correspond to top positive match candidates for the reference image in (a), with x-coordinates representing the top negative match score.

Several displays comparing the scores of positive and negative match candidates are shown in Figure 11. To compare the scores of all positive match candidates against the negative match candidates (for the reference image in Figure 11-(a)), we display histograms of the similarity score distributions for positive match candidates as a function of exclusion radius (blue), to be compared to the histogram of negative candidates scores (orange) displayed at right in Figure 11-(b). The scores of the top positive matches are connected by a blue dotted line, corresponding to the images displayed in sequence in the image panel above (Figure 11-(a)). The score of the best matching negative match candidate is marked using a red horizontal line. A matching error occurs when the score of the top positive match falls below that of the top

negative match. While the ShapeY matching task focuses on the top positive and negative match scores, the distributions show that many negative matches can score higher than many positive match candidates. This means that the network's embeddings are not well-separated, and ShapeY's exclusion mechanism reveals the structure by systematically removing positive examples.

In Figure 11-(c), we present a scatter plot of the top positive and negative match candidate scores for all matching tasks in the **pr** series. The color gradient indicates the exclusion radius for each task. The red dotted line represents $y = x$; points below this line indicate match failures. This scatter plot, which aggregates data from all matching tasks, allows us to observe how quickly positive match scores decay relative to the top negative match. A faster decay suggests greater overlap in the embedding space. We will use this scatter plot to compare DNs with different ShapeY performance levels for further analysis in a later section.

### 3.5 Error examples provide a qualitative view of the embedding space

The numerical results shown in Figure 7 provide a quantitative summary of the shape representation capabilities of a vision system, and especially the ability to tolerate 3D viewpoint variation. Our approach to nearest-neighbor matching with exclusions can also provide a *qualitative* measure of the "tangledness" of the embedding space, by analyzing shape match failures.

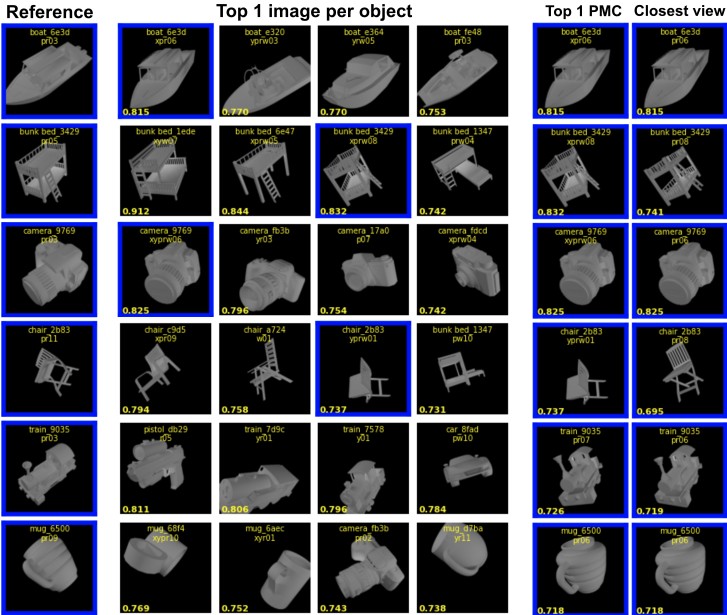

Figure 12: Example image panel displaying six rows of: (left) the reference image; (middle) rank-ordered match candidates (top 1 per object) in the descending order of cosine similarity (according to ResNet50); and (right) top 1 positive match candidate and the closest viewpoint within the same series available as positive match candidate after exclusion. We apply $r_e = 2$ in **pr** for all six examples. In each image, we display the image's object name and series index on top, and its similarity to the reference on the bottom left. We highlight the positive match candidates in blue.

Figure 12 displays an image panel with six example rows displaying the reference image on the left, the rank-ordered match candidates (top 1 image per object) in the middle, followed by the best matching positive match candidate and an image that is rendered in the closest viewpoint to the reference at the specified exclusion radius ($r_e = 2$ in **pr**). We highlight the boundaries of the positive match candidate images to help visually inspect whether the tested network (ResNet50) is correct or not. If the second column of the panel is not highlighted in blue, that means the network incorrectly matched in an object test.

Match failures are particularly informative regarding the quality of the embedding: when a view of a reference object is found to be very close to a view of even one other object of very different shape, it is likely that

the reference view is close to a large number of other very different shapes as well, whose discovery depends mainly on having a sufficient number of distractors in the view database. For example, if ResNet50 puts a view of a pistol (row 5, second column in Figure 12) closer than a view of a train (row 5, top1 PMC) to another slightly rotated view of a train (row 5, reference image), it is very likely to mistakenly match the train to many other objects of different shapes.

### 3.6 ResNet50's nearest-neighbor matching failures do not stem from out-of-distribution problem

Table 2: Linear probing results

|  | Accuracy (Stdev) | Chance | Cohen's $\kappa$ |
|---|---|---|---|
| Category-as-class (20 output units) | 0.8633 (0.0010) | 0.05 | 0.8561 |
| Object-as-class (200 output units) | 0.7913 (0.0015) | 0.005 | 0.7903 |
| ImageNet1k | 0.808 | 0.001 | 0.8078 |

Why does the pre-trained ResNet50 perform so poorly on our benchmark? One potential explanation is that the images are out-of-distribution (OOD). In other words, the network may struggle to visually parse unfamiliar grayscale-shaded objects. Here, we hypothesize that this deficiency could occur at the embedding level and investigate this possibility by freezing the embedding and training a single layer of linear weights between the embedding vector (dim=2048) and either 20 (object category as class) or 200 (individual object as class) category units—a process referred to as linear probing (Chen et al., 2020). If the embedding vectors are truly deficient and unable to handle these supposedly OOD images, a single layer of weights will fail to correctly activate the output category units. Conversely, if the embedding can effectively represent ShapeY images, linear probing would enable accurate classification into respective categories.

We train the linear layer for 10 epochs using an SGD optimizer (learning rate = 0.001, momentum = 0.9) and repeated the experiment 5 times. Table 2 summarizes the linear probing accuracies. The top-1 linear probing accuracies for classifying into object categories (86.3%) and individual object exemplars (79.9%) are both far above chance levels. These results are comparable to the model's accuracy on ImageNet1k (80.8%), even if we consider that we train with a smaller number of categories (chance adjusted accuracies, or Cohen's Kappa in Table 2, are also on par with the model's ImageNet1k accuracy). Therefore, we conclude that the ResNet50 pre-trained on ImageNet1k adequately represents the ShapeY images and does not face a serious OOD problem.

Additionally, the discrepancy between the pre-trained ResNet50's performance on our nearest-neighbor matching task and its classification accuracies suggests that the ShapeY matching task is inherently more challenging. This difference—where k-NN performance establishes a lower bound for linear probing results—has been observed experimentally in a previous study (Oquab et al., 2024, see caption for Table 1). Geometrically, while images rendered from the same object or object category only need to reside within loosely defined clusters to support satisfactory classification performance, to enable accurate nearest-neighbor matching, images must be organized into well-separated "pure" clusters. In other words, strong performance on the ShapeY benchmark implies a capacity for strong classification, while the converse may not be true.

### 3.7 Self-supervised ResNet50 performs worse than supervised counterpart

Our nearest-neighbor-based matching task is similar to contrastive training objectives, where we aim for slightly augmented images to be adjacent in the embedding space of an object recognition system. Therefore, we hypothesize that a ResNet50 backbone—architecturally identical to our supervised baseline, but pre-trained with a self-supervised objective instead of supervised cross-entropy on ImageNet labels—may perform better on our benchmark. We evaluate ResNet50 backbones pre-trained using various self-supervised objectives (SimCLR (Chen et al., 2020), SimCLR-DCL (Yeh et al., 2022), SimCLR-DCLW (Yeh et al., 2022), SwAV (Caron et al., 2021a), VICReg (Bardes et al., 2022), MoCo (He et al., 2020), BYOL (Grill et al., 2020),

Barlow Twins (Zbontar et al., 2021), DINO (Caron et al., 2021b), and DirectCLR (Jing et al., 2022)) to see if they perform better on our benchmark. We obtain their pre-trained weights (on ImageNet1k) from *LightlySSL*.

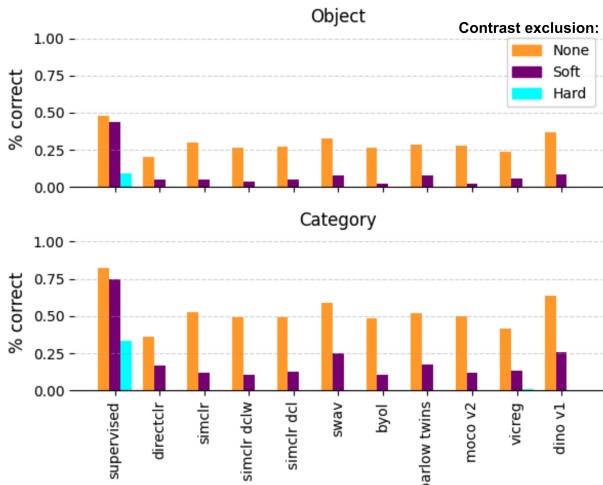

Figure 13: Nearest-neighbor matching accuracies ($r_e = 2$ in **pr**) of several pre-trained ResNet50s on various self-supervised objectives (higher the better). All of the self-supervised ResNet50s performed significantly worse than their supervised counterpart. In addition, the self-supervised networks are significantly affected by a simple background color change (purple and cyan), scoring mostly 0% on the hard contrast-reversal task.

Figure 13 shows the object and category-level matching accuracies at $r_e = 2$ in **pr** (higher is better). Contrary to our hypothesis, all the self-supervised methods perform worse than the supervised ResNet50. The best-performing self-supervised method is DINO, which still underperforms the supervised ResNet50 by nearly 20% in the object-level matching task. SimCLR, previously noted for being more shape-biased (Geirhos et al., 2020b), also significantly underperforms compared to the supervised ResNet50.

While viewing angle modifications may not be part of the augmentations these self-supervised methods are trained to counteract, it is surprising to see that the self-supervised methods are more severely affected by a simple background change. Almost all of them scored 0% accuracy on the hard contrast exclusion task (cyan bars, Figure 13). Note that a background change is more similar to the image augmentations (which are mostly post-processing done on the images) they encounter during training.

### 3.8 Can fine-tuning a network with ShapeY image set teach it a generalizable sense of 3D shape?

While we have shown that the embeddings of pre-trained ResNets, supervised or self-supervised, do not display shape understanding despite the images being adequately represented, there still remains a question about whether introducing image variations that naturally occur with 3D transformations, instead of just post-processed random image augmentations, helps to achieve a generalizable shape understanding in DNs. To answer this question, we fine-tune the pre-trained ResNet50 using our images.

Fine-tuning is beneficial in that we can align the DN to our images, eliminating the OOD concern completely. However, the downside of training with our images is that it becomes harder to test whether the trained network's shape understanding generalizes to novel objects. To mitigate this, our strategy is to fine-tune the ResNet50 with a subset of the ShapeY objects and run the ShapeY benchmark on the left-out set of objects to test generalization to unseen objects. If the improvement in the ShapeY benchmark transfers to a left-out set of objects, the network can be claimed to have aquired some shape understanding capability, though limited to our simple and controlled image set.

We fine-tune the network in three different ways using our image set:

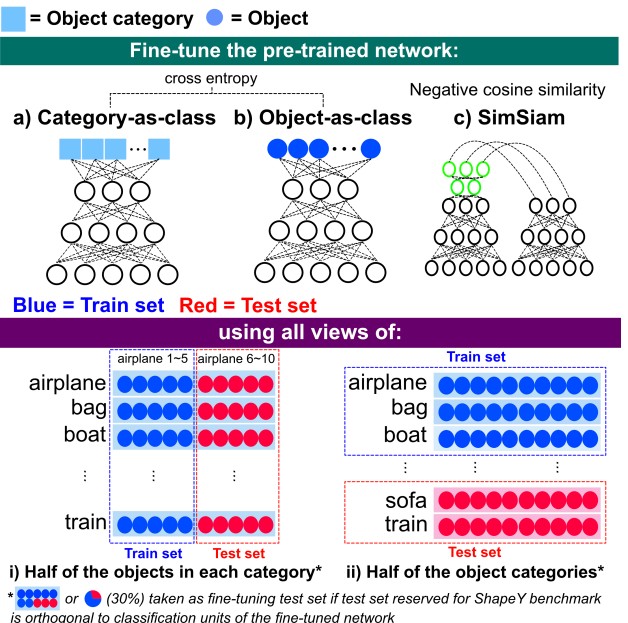

Figure 14: Schematic description of different fine-tuning experiments. We fine-tune a pre-trained ResNet using a cross-entropy objective, taking either object categories (**a**) or objects (**b**) as output class units, or a negative cosine similarity objective (**c**) that maps two adjacent views of an object closer in the embedding space. To test whether the improvement on ShapeY benchmark, if any, generalizes to an unseen object, we leave out either half of the object in each category (i) or half of object categories (ii) as a test set.

a) Fine-tune the network on classification objective using ShapeY object categories as class

b) Fine-tune the network on classification objective using ShapeY objects as class

c) Fine-tune the network on negative cosine similarity, maximizing similarity between embedding vectors of the two views of the same object (SimSiam)

In **(a)**, we test whether providing rough supervision on the object category level enhances the network's shape understanding capability. In **(b)**, we test whether more specific supervision telling the network to separately classify each object would help. Finally, in **(c)**, we tested whether self-supervision that tells the network to map two adjacent views (with an exclusion radius of at least 2) to a similar embedding vector can make the network understand 3D shape.

For each of the fine-tuning cases listed above, we train the network using:

i half of the objects (5 out of 10) in each category

ii half of the object categories (10 out of 20)

For example, the network that is fine-tuned using category-as-class output units (**a**) is trained to classify either all of the views of the 5 objects in each category into 20 categories (**a**-i) or all of the views of all 10 objects in half of the object categories into 10 output category units (**a**-ii). Such test set divisions are done so that we can quantify whether the shape understanding capability generalizes to unseen objects (i) and unseen object categories (ii). We also observe whether the network's performance on ShapeY's nearest-neighbor classification improves on the training set.

For the case when the test set, or images that are concealed during training, only contains objects that do not have corresponding classification output units when fine-tuning (for cases **a** and **b**), we withhold

an additional 30% of the training set to create a classification test set. For instance, when fine-tuning the network with 10 categories as class outputs (**a**-ii), because the other 10 categories cannot be used to test the network's classification accuracy, we additionally hide 3 out of 10 objects in each of the 10 object categories in the train set to compute the test classification accuracy. Similarly, when the network is fine-tuned with 100 objects as class outputs (**b**-i and **b**-ii), 30% of the views of each object are additionally concealed during training and used as a test set to compute the network's classification accuracy for unseen views. Figure 14 provides a schematic overview of these three fine-tuning objectives and the two held-out splits.

We fine-tune using the SGD optimizer with a learning rate of $5e^{-4}$ and momentum of 0.9, and train the network for 20 epochs (cases **a** and **b**). For SimSiam (**c**), we again use the SGD optimizer with a learning rate of 0.0025, which is adjusted according to the batch size used (64), momentum of 0.9, and weight decay of $5e^{-4}$. We use two fully connected layers (input dim=2048, hidden dim=1024, output dim=2048) as the projection layer in SimSiam structure. During training, we present two views of the same object that are along the same, or closest, ShapeY series but are at least $r_e = 2$ away.

Figure 15 below summarizes the results. We fine-tune each case 5 times with randomly selected train and test objects (i and ii) and show the average achieved nearest-neighbor accuracies ($r_e = 2$ in **pr**) and their 95% confidence intervals (CI).

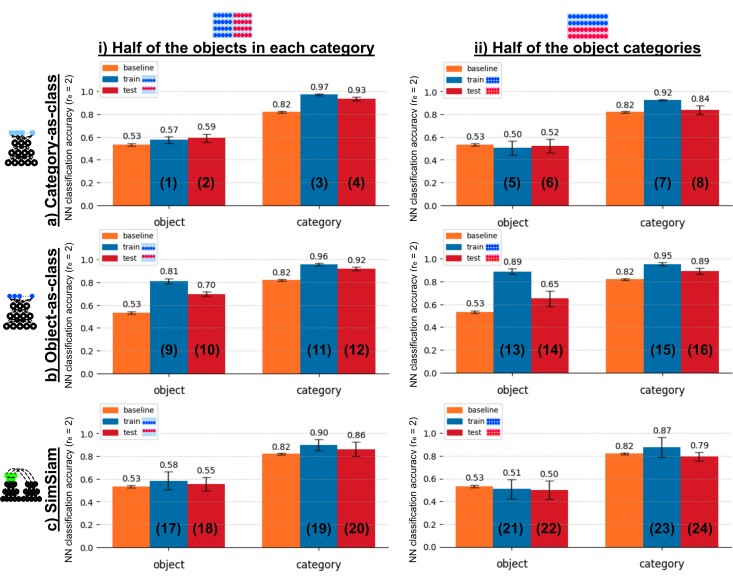

Figure 15: Fine-tuning results reporting nearest-neighbor matching accuracies at $r_e = 2$ in **pr** (higher the better). Orange displays the pre-trained ResNet50's accuracies, blue displays the fine-tuned networks' accuracies on trained 3D objects, and red shows accuracies on held-out objects. We conduct 5 fine-tuning experiments with randomly pick train and test objects each time, and show the average accuracies achieved (number written on top of each bar) along with the 95% confidence intervals (error bars).

The fine-tuning experiments yielded several key insights. First, supervision through cross-entropy can disentangle the embedding space for trained objects but fails to generalize well to unseen ones. For instance, when the network is fine-tuned using object-category-as-class (**a**), its nearest-neighbor classification accuracy on the object categories improved to 97% and 93% , which are higher than the baseline (orange, 82%) with statistical significance, for train and test objects (**a**-i, category) respectively (*(3)* and *(4)* in Figure 15). Here, because we take half of the objects in each category as test sets, the network has seen example objects within each category. On the contrary, when the network is tested on unseen object categories, the improvement disappears (84% with CIs overlapping with the baseline, *(8)* in Figure 15).

Similarly, fine-tuning the network using object-as-class, we observe larger improvements in the nearest-neighbor classification accuracy for objects that are in the train set, and modest improvements in the objects in the test set. Object level accuracy improves to 81% when trained with half of the objects in each object

category (**b**-i, train set, *(9)* in Figure 15) and to 89% when trained with all objects in the trained categories (b-ii, train set, *(13)* in Figure 15) from 53% accuracy of the baseline. On the other hand, the accuracies reduce to 70% (**b**-i, test set, *(10)* in Figure 15) and 65% (**b**-ii, test set, *(14)* in Figure 15) when tested on unseen objects, which are still statistically higher than the baseline accuracy. This implies that providing more granular supervision on object-level is more effective in arriving at a generalizable shape understanding than supervision on object-category-level.

Second, fine-tuning with SimSiam, which enforces the embedding vectors of viewpoint-augmented images to be similar, only leads to disentanglement at the object-category level, rather than developing object-specific encoding (**c**-i and **c**-ii). The observed improvement trends are identical to fine-tuning with category-as-class (**a**) on the ShapeY benchmark for the network fine-tuned with SimSiam, where the category-level accuracies only improve on the trained sets (*(19)* in Figure 15) and the object-level accuracies do not improve at all (*(17)*, *(18)*, *(21)*, and *(22)*). This is surprising, as we hypothesize that providing supervision to map two views closely in embedding space would improve object-level accuracies. We conclude that simply telling the network that the two views of the same object should be similar in the embedding space does not make the network suddenly understand the 3D shape.

Finally, though fine-tuning makes the network perform better in the viewpoint exclusion tasks, it causes the network to *completely* fail on the contrast-exclusion test. All of the fine-tuned networks start to show near 100% error rates with simple background inversion (hard contrast reversal test) even without any viewpoint exclusion. Though we can include background-inverted images during training, this result signals that the network's performance will degrade and fail on some other variation that is not included in the training set. Therefore, we conclude that the ShapeY image set should remain a validation set to test shape understanding capabilities, rather than being included in the train set.

### 3.9 Can other network architectures support better shape understanding?

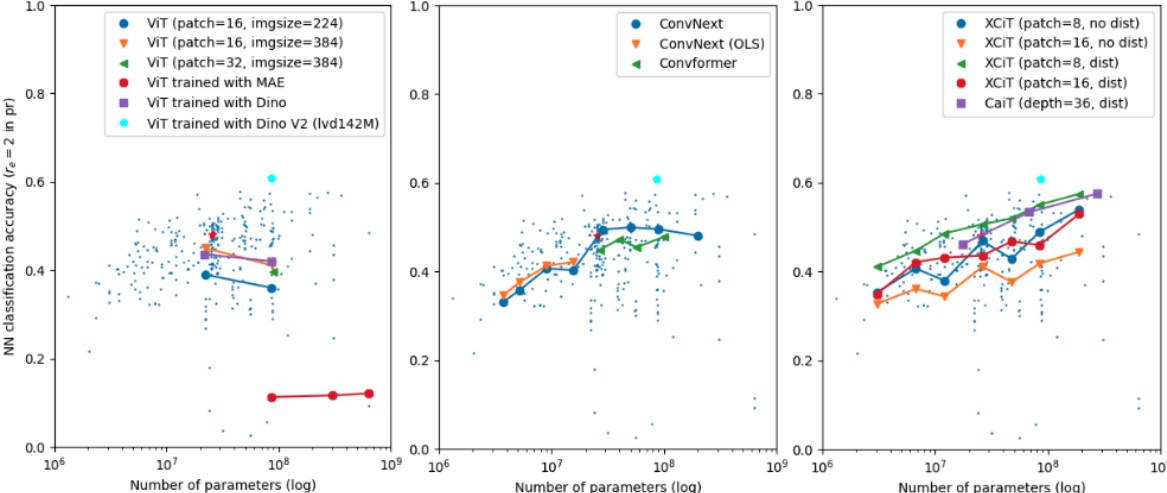

Figure 16: Nearest-neighbor matching accuracies ($r_e = 2$ in **pr**) plotted against number of parameters of 321 different DN architectures pre-trained on ImageNet1k with an exception of a ViT trained with DINOv2, where authors curated a larger dataset themselves (142 million images) (Oquab et al., 2024) (weights taken from *timm* (Wightman et al., 2021)). We plot the scaling behaviors of three notable architectures: (left) ViT (Dosovitskiy et al., 2021); (middle) ConvNext (Liu et al., 2022) and Convformer (Gu et al., 2022); (right) XCiT (El-Nouby et al., 2021) and CaiT (Touvron et al., 2021). ResNet50 is marked using a red star in all panels, and DINOv2, which is the best performing network, is marked with a cyan colored pentagon. ViT and convolution-based networks do not improve on the ShapeY matching task with scale, while the transformer-variants XCiT and CaiT improves with scale.

Can a more modern DN architecture better perform on the ShapeY benchmark? We sweep 321 different pre-trained networks taken from the *timm* library (Wightman et al., 2021) to test a variety of DN architectures. Several of those network weights are similar in architecture, with differences in scale or in training objectives. We limit to networks pre-trained on ImageNet1k, with DINOv2 (Oquab et al., 2024) being a notable exception where it is pre-trained on a larger image set ($\sim$ 142 million images) that the authors curated themselves. We include this as it outperforms all 321 tested networks on the ShapeY benchmark.

Figure 16 plots the matching accuracies ($r_e = 2$ in **pr**) of all of the tested pre-trained networks against their number of parameters in logarithmic scale on the x axis. We mark the pre-trained ResNet50 with a red star and the ViT trained with DINOv2 objective with a cyan-colored pentagon in all of the panels. DINOv2, the best performing network, shows 62% object-level matching accuracy with an exclusion radius of 2 in **pr**. To visualize the effect of scale, training objectives, and some model parameters for some notable architectures, we group similar architectures in each of the subplots.

Different scaling trends emerge for these groups. From left to right, Figure 16 displays scaling trends for variants of Vision Transformers (ViTs) (Dosovitskiy et al., 2021), convolution-based architectures (ConvNeXt (Liu et al., 2022) and ConvFormer (Gu et al., 2022), which combines CNN with transformer layers), and other transformer-based architectures (CaiT (Touvron et al., 2021) and XCiT (El-Nouby et al., 2021)). We first see not only that ViTs' scores deteriorate with scale, but also generally perform worse than ResNet50 on our benchmark. On the other hand, the middle plot shows that the ShapeY score of the convolution-based architecture (ConvNeXt) improves with scale until it plateaus and degrades. In contrast, the right plot shows that other transformer-based networks (CaiT and XCiT) improve as they become larger. These results show that scaling can work favorably for developing a generalizable shape understanding only in some network architectures. However, even for the transformer-based networks, we only observe <10% improvement in the matching accuracy for going 10x larger with our modestly-sized (200) object set.

We additionally observe how changing model parameters as well as the training objective affect the ShapeY performance of transformer-based networks. ViTs trained with masked autoencoder (He et al., 2021) fails completely (red circle, left, Figure 16). Random patch erasures introduced during training for reconstruction could be attributable to this behavior because most of the shape information seems to be discarded with such image modification (see figures 2-4 in He et al. (2021) for examples). On the other hand, distillation objectives seem to guide the network to perform as well or sometimes better than the supervised ones. For example, DINO (purple square, left, Figure 16) performs on par with the supervised ViTs (orange triangle). In XCiTs, distillation using a pre-trained CNN teacher boosts their matching accuracies by $\sim$5% on average (right, Figure 16). DINOv2 (cyan pentagon, Figure 16) also combines patch-level distillation with global-level distillation and outperforms every tested network, although it is trained on a larger dataset.

### 3.10 Poor performing systems suffer from dimensional collapse

What could be the cause of the observed performance variations in the ShapeY across different tested systems? Figure 17 presents the top 1 negative and positive match candidate score pair scatter plots for six selected networks. We organize the figure into columns to compare similar architectures with slightly different training parameters. The top row displays the worse-performing versions out of the two. A consistent pattern emerges in the scatter plots, where the score pairs are more tightly clustered in the top right corner for the less performing systems. In other words, both the positive and negative match candidates exhibit high similarities to the reference in the embedding spaces of shape-insensitive systems. For some networks like ViT trained with masked autoencoder (MAE), the clustered pattern signals potential dimensional collapse leading to inadequate representation of ShapeY images. This makes sense intuitively since the MAE focuses on reconstruction, which may have caused the network to learn features for interpolation which aren't useful for distinguishing based on semantics or 3D shape.

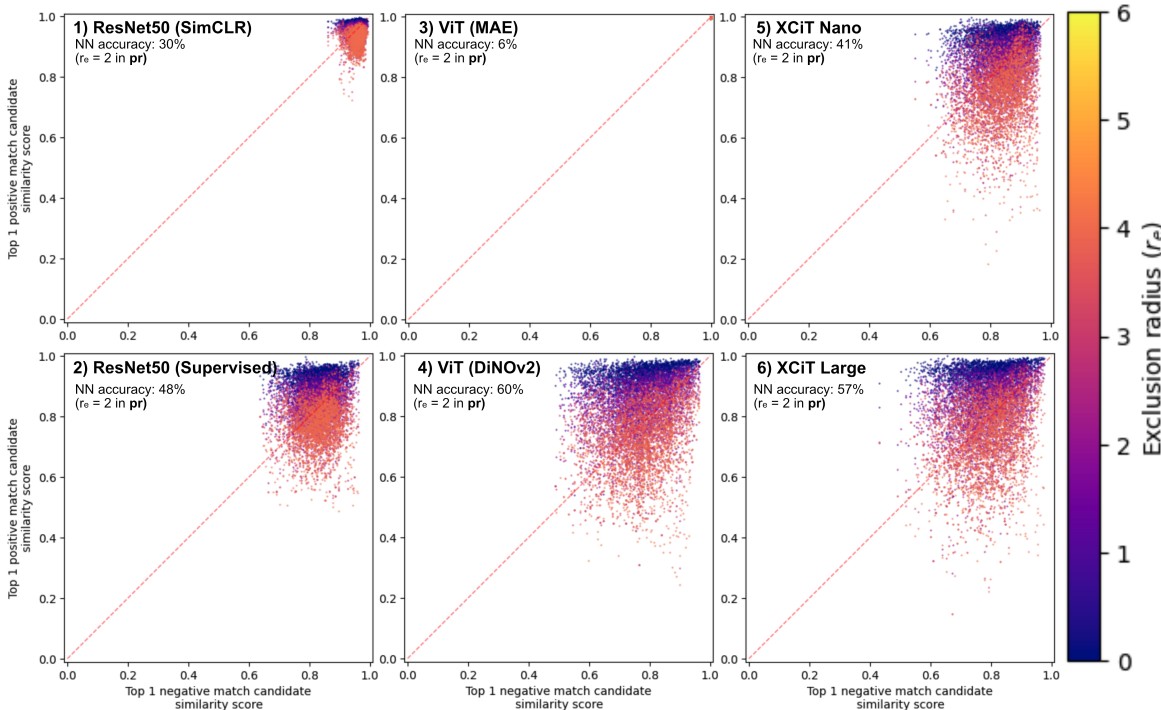

Figure 17: Top 1 negative vs positive match candidate score pair scatter plots of selected networks (**pr** series). **(1&2)** ResNet50 trained with SimCLR vs supervised objectives. **(3&4)** ViTs trained with masked autoencoder (MAE) vs DINOv2. **(5&6)** XCiTs with smaller (Nano) vs larger (Large) number of parameters.

## 4 Discussion

### 4.1 Beyond traditional benchmarks: towards principled evaluation of robustness in object recognition systems

Human visual object recognition relies heavily on shape due to its diagnostic value and consistency across changes in viewpoint, surface properties, lighting, and atmospheric conditions. Traditional object recognition benchmarks like ImageNet do not emphasize shape, often classifying images with widely varying shapes into the same category (Figure 18). Although success in such benchmarks can indicate a system's generalization capability, studies have shown that DNs trained on traditional OR image sets often learn category-irrelevant features (Geirhos et al., 2020a) and perform differently when shown covariate-shifted images (Hendrycks & Dietterich, 2019; Wang et al., 2019; Yang et al., 2024). While we can continue to explore where OR systems fail to generalize, there is a clear need for a principled evaluation metric that guides the creation of a robust and human-aligned vision system.

Based on this need, we have introduced ShapeY, a shape-based benchmarking system. ShapeY's design is based on the idea that an object recognition system should consistently judge that a view of an object is most similar to another view of the same object, regardless of changes in the object's surface properties, 3D pose, or viewing conditions, compared to any other view of any other object under any viewing conditions. This simple premise forms the basis for measuring shape-based object recognition capability using a nearest-neighbor matching task.

Geometrically, ShapeY helps to characterize the fine-grained structure of an OR system's embedding space. For a network to perform well on classification tasks, its embedding space only needs to be coarsely organized into categorical structures before the fully connected layer. This means that a network's embedding, which performs well on traditional OR benchmarks, can still be severely entangled at a local level. ShapeY reveals this local entanglement by gradually removing nearby positive samples, controlling task difficulty,

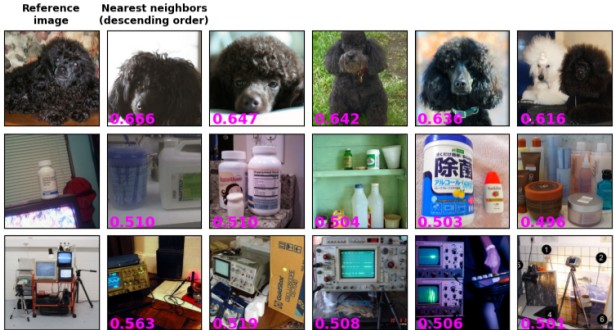

Figure 18: ImageNet images (subset of 100,000), rank-ordered by correlation values (magenta) to the reference (left) in the embedding space of a pre-trained ResNet50. In some cases, some global shape similarity is evident in addition to color/texture similarity (top row), while in most cases, shape-based matching is clearly de-emphasized (other rows).

and repeatedly measuring nearest-neighbor classification accuracy (Figure 1). Task difficulty is adjusted using viewpoint exclusions, which remove physically close 3D views from the positive match candidates, and appearance exclusions, which force matching across changes in non-shape attributes. Degradation in nearest-neighbor classification performance is thus interpreted as the system's sensitivity to the required exclusions.

Unlike traditional OR benchmarks, ShapeY offers a suite of quantitative and qualitative reports to evaluate OR system performance: (1) object and category error rates, segmented by viewpoint transformation type and magnitude, with and without appearance exclusions (Figures 7, 9); (2) similarity score decay as the query view pose changes (Figure 10); (3) image grids displaying top matches for reference views, enabling visual assessment of matching errors (Figure 12); (4) match rank histograms, showing the best positive match candidate ranks across all reference images; and (5) distractor encroachment histograms for single reference images, illustrating similarity score distributions for positive and negative match candidates and highlighting distractors that outscore the best-matching positive candidates at each exclusion distance, causing matching errors. In summary, ShapeY's comprehensive set of numerical and graphical reports provides a detailed overview of the capabilities and weaknesses of an OR system across a spectrum of task difficulties.

## 4.2 The poor performance of ResNet50 and many DN-based OR systems may stem from their architectures

Even with a modestly sized image set (68k images, 200 objects), architectures like ResNet50, which perform well on the ImageNet-1k benchmark, struggle on the ShapeY benchmark, particularly with depth rotations. In retrospect, this poor performance is not entirely surprising, as many deep network-based OR systems are known to rely on non-shape cues for classification (Baker et al., 2018; Geirhos et al., 2019). Interestingly, focusing on image contours alone does not improve performance on our matching task: ResNet50 trained to be more shape-biased using Stylized ImageNet (Geirhos et al., 2019) performs worse than its ImageNet-trained counterpart (27% accuracy, $r_e$=2 in **pr**). Clearly, shape is more than just the image contour itself, and forcing ResNet50 to be contour-biased does not help.

We investigated the underlying cause of the lack of shape understanding in the supervised ResNet50. We hypothesized that the high error rates in the ShapeY task could be due to our grayscale object images being out-of-distribution (OOD) for the pre-trained models, or perhaps a lack of experience with our object classes in the pre-trained embedding space. This hypothesis was ruled out, as linear probing on the pre-trained embedding performed nearly as well as it did for ImageNet-1k when classifying our images. Thus, we found that DN-based vision architectures can handle these grayscale, textureless images and ShapeY object categories, but they struggle with judging similarity across 3D viewpoint changes and even with trivial appearance changes, such as altering the background color—analogous to placing the object on a different tabletop.

We also experimented with fine-tuning ResNet50, both in supervised and self-supervised manners, to see if including 3D view variations could improve shape understanding. While we observed some minor improvements in ShapeY scores that transferred to unseen objects, the accuracy dropped to near zero when tested with contrast-exclusion (matching across background color). This raises questions about whether the network learned generalizable shape features. We thus conclude that conventional networks face significant challenges in handling shape independently of 3D viewpoint and other non-shape appearance changes, indicating a deeper, perhaps architectural, problem with shape understanding in DN-based systems.

### 4.3 OCD errors persist in the best performing DN architecture.

Testing approximately 300 pre-trained object recognition systems, we found no single solution for enhancing shape understanding in deep network architectures. In some architectures, such as XCiT, scaling the network parameters exponentially leads to linear improvement in ShapeY scores. However, in other architectures, such as ViT or convolution-based DNs, scaling does not seem to help. The training objective also appears to affect ShapeY scores: ViTs trained with a masked-autoencoder reconstruction objective perform the worst. The best-performing network, DINOv2 (Oquab et al., 2024), utilized a larger curated dataset, a ViT-based architecture, and distillation-based self-supervision. While it is unclear which specific factors led to the improvement, it is evident that ShapeY provides a different yet crucial perspective not offered by traditional classification benchmarks.

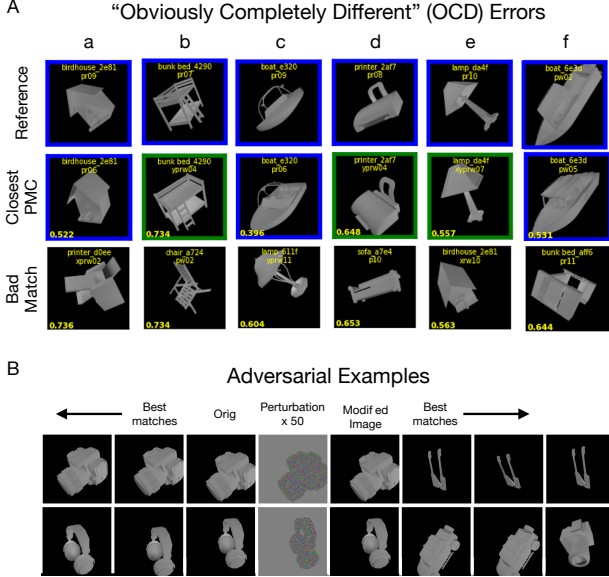

Figure 19: **(a)** Nearest-neighbor matching errors for DINOv2 ($r_e = 2$ in **pr**). Examples show clear mismatches across categories that humans would not make (OCD errors). **(b)** Adversarial examples of ShapeY images altered to resemble a different object category in the embedding space.

Despite its strong performance, DINOv2's embedding space still exhibits significant entanglement. Figure 19 highlights obviously completely different matching errors, which are clear mismatches that humans would never make. This indicates that even in the best-performing network, images in the embedding space remain closely surrounded by distractors, undermining its robustness. To further probe this, we created adversarial examples by deliberately pushing an image's embedding toward another object's embedding (Figure 19-**b**). These examples, while artificial, underscore the vulnerability of even state-of-the-art systems when faced with a larger or more diverse set of distractors.

Currently, ShapeY includes only 200 objects across 20 categories, a modest subset of the roughly 1,000 basic-level categories humans recognize (Biederman, 1987). As object recognition systems continue to improve, benchmarks like ShapeY must scale and evolve to remain effective. Expanding ShapeY to include a broader

range of objects, categories, and non-shape variations—such as scale changes, textured objects, and non-rigid transformations—will be critical for uncovering increasingly subtle failure modes and driving the development of truly robust, human-aligned vision systems.

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
