# OpenReview forum: "ShapeY: A Principled Framework for Measuring Shape Recognition Capacity via Nearest-Neighbor Matching"
_TMLR — Under review for TMLR_

### Review · Reviewer_gQE5 · 2026-05-20

**Summary Of Contributions:**

This paper introduces ShapeY, a controlled benchmark for evaluating shape-based object recognition in visual models. The benchmark consists of 68,200 grayscale rendered images from 200 3D objects, organized into 20 basic-level categories with 10 instances each. Each object is rendered from multiple controlled viewpoints, including pitch, yaw, roll, and translations, producing 341 views per object. The authors use a nearest-neighbor matching task in model embedding space to test whether different views of the same object are closer to each other than views of other objects. The benchmark also includes viewpoint exclusion and appearance exclusion settings to test whether models can match objects across nontrivial 3D pose and background/appearance changes.

The paper argues that standard benchmarks such as ImageNet do not directly measure shape-based recognition because ImageNet categories often contain large shape variation, cluttered scenes, and many non-shape cues such as texture, color, and context. ShapeY is proposed as a more principled benchmark for testing whether object recognition systems organize their embeddings according to 3D shape similarity. The experiments evaluate a ResNet50, several self-supervised ResNet50 models, fine-tuned variants, and 321 pretrained networks. The main finding is that many strong ImageNet models perform poorly on ShapeY, especially under depth rotations and appearance changes, suggesting that their embedding spaces remain locally entangled with respect to 3D shape.

A major strength of the benchmark is its controlled design. By using rendered 3D objects, the authors can systematically vary viewpoint and appearance while holding object identity fixed. This makes the benchmark more diagnostic than natural-image classification datasets, where viewpoint, background, texture, lighting, and object identity are heavily confounded.

**Audience:**

Yes

**Audience Explanation:**

The construction of the vision dataset or benchmark is the critival problem in the community of machine learning and computer vision.

**Claims And Evidence:**

No

**Claims Explanation:**

The paper is mainly motivated by the difference of "shape" and "texture" in the vision dataset. However, the connection between the motivation and the construction of shapeY is not clearly illustrated and not convincing. The decoupling of "shape" and "texture" is a partially good motivation, but it is not the first principle of the construction of the vision dataset or vision benchmark. Moreover, even if the decoupling is achieved by shapeY, most researchers cannot replace ImageNet with shapeY. ImageNet is still the common validation benchmark. Please refer to the "Requested Changes".

**Requested Changes:**

1. **The connection between “shape vs. texture bias” and the proposed multi-view benchmark should be clarified.**

The paper motivates ShapeY by discussing the difference between shape and texture cues in datasets such as ImageNet. However, the proposed benchmark mainly evaluates viewpoint-invariant 3D object matching, not shape-vs-texture cue conflict in the usual sense. These are related but distinct problems.

For example, a model could be texture-biased on ImageNet-style cue-conflict images but still perform reasonably well on some multi-view matching tasks. Conversely, a model could be less texture-biased but still fail to recognize the same object across large 3D rotations. ShapeY removes much of the texture information by using grayscale, textureless objects, so it is not directly testing whether a model prefers texture over shape when both cues are available. Instead, it tests whether the model’s embedding space preserves object identity across controlled 3D transformations.

This does not invalidate the benchmark, but the paper should be more precise about its scope. I would recommend that the authors frame ShapeY primarily as a benchmark for 3D shape-based viewpoint invariance, rather than as a general solution to the shape-vs-texture bias problem.

2. **The advantage of "shape"**
The paper empahsizes the advantage of "shape" in the object recognition system. Is this a common sense in the community? I suggest that authors should list more evidences to support this claim.

---

> ### Author Response · Authors · 2026-07-04
>
> # Overall comment
> **The paper is mainly motivated by the difference of "shape" and "texture" in the vision dataset. However, the connection between the motivation and the construction of shapeY is not clearly illustrated and not convincing. The decoupling of "shape" and "texture" is a partially good motivation, but it is not the first principle of the construction of the vision dataset or vision benchmark.**
>
> Our comment:
> We thank the reviewer for this comment, and acknowledge that our motivation was not well expressed in our original submission. To clarify, our work is not motivated by, and does not explore, shape-versus-texture bias in vision networks or datasets. Rather, the design of the ShapeY benchmark was driven by the fact that in humans, the ability to recognize and categorize objects, despite variations in 3D viewpoint, lighting, surface properties, optical conditions, etc., is known to depend primarily on shape.  In contrast, state-of-the-art artificial vision systems based on deep networks rely on an intricate mixture of shape, texture and color cues that defies simple characterization.  ShapeY isolates the shape representing capabilities of machine vision systems, and in so doing, can help track progress of artificial vision systems along the dimension that aligns most closely with human recognition behavior.
>
> We have updated the abstract to more accurately reflect this motivation.
>
> **Moreover, even if the decoupling is achieved by shapeY, most researchers cannot replace ImageNet with shapeY. ImageNet is still the common validation benchmark. Please refer to the "Requested Changes".**
>
> Our comment:
> We do not argue that ShapeY should replace ImageNet, and we acknowledge ImageNet’s value as a training and validation resource. Our discussion of ImageNet concerns its suitability as an evaluation benchmark for shape-based recognition specifically. For the reasons discussed in the revised Introduction (pages 2-3), success on ImageNet requires a mix of several capabilities in uncontrolled amounts.  ShapeY, in contrast, provides a purer measure of shape-based recognition capability, by measuring a system’s ability to match same-object views across controlled 3D viewpoint changes, and despite optional appearance (i.e. non-shape) changes.   ShapeY is meant to complement, not replace, ImageNet-style benchmarks.
> The same division exists in language modeling, where models are pretrained on massive, heterogeneous text corpora but evaluated on targeted benchmarks (e.g., MMLU, GSM8K) designed to cleanly probe a single capability.

---

> ### Author Response · Authors · 2026-07-04
>
> ## Comment 1
> **The connection between “shape vs. texture bias” and the proposed multi-view benchmark should be clarified.**
> We agree
>
> **The paper motivates ShapeY by discussing the difference between shape and texture cues in datasets such as ImageNet.**
> We have changed this in the abstract and introduction to make the distinction more clear.
>
> **However, the proposed benchmark mainly evaluates viewpoint-invariant 3D object matching, not shape-vs-texture cue conflict in the usual sense. These are related but distinct problems.**
> This is correct.
>
> **For example, a model could be texture-biased on ImageNet-style cue-conflict images but still perform reasonably well on some multi-view matching tasks.  Conversely, a model could be less texture-biased but still fail to recognize the same object across large 3D rotations. ShapeY removes much of the texture information by using grayscale, textureless objects, so it is not directly testing whether a model prefers texture over shape when both cues are available. Instead, it tests whether the model’s embedding space preserves object identity across controlled 3D transformations.**
> We believe this is probably true for the best performing models we tested.
>
> **This does not invalidate the benchmark, but the paper should be more precise about its scope. I would recommend that the authors frame ShapeY primarily as a benchmark for 3D shape-based viewpoint invariance, rather than as a general solution to the shape-vs-texture bias problem.**
> We agree with this comment.  Accordingly, we have rewritten the abstract in such a way as to clarify the scope.  We also minimized use of the term “shape-bias” throughout the paper, and added sentences that explicitly distinguish our work from the existing shape-bias literature (discussed on pages 2-3).

---

> ### Author Response · Authors · 2026-07-04
>
> ## Comment 2
> **The paper emphasizes the advantage of "shape" in the object recognition system. Is this a common sense in the community?  I suggest that authors should list more evidences to support this claim.**
>
> There are wide ranging views in the community regarding the importance of shape for recognition.  There are those who have worked to increase the “shape bias” of DN-based vision systems (see references in the paper), and those that view the high performance of conventional networks on benchmarks such as ImageNet as a validation of whatever inscrutable combination of shape, color, texture, lighting, atmospheric and context cues those networks represent about images.
> What is less controversial is that human similarity judgements in comparing objects and scenes depend primarily on the similarity of shape.
>
> The centrality of shape to human object recognition is a well-established finding in the vision science literature. In the introduction, we cite several papers including Biederman (1987), Hoffman (1998), Grill-Spector et al. (2001), and Kourtzi (2001) — which discuss the evidence that shape is the primary cue for basic-level categorization in humans. We also explicitly discuss the complementary roles of shape and non-shape cues: texture and color are important for subordinate-level distinctions (e.g., distinguishing a robin from a sparrow), but shape remains the essential basis for basic-level categorization (e.g., recognizing something as a bird) in human vision.

---

### Review · Reviewer_v8qF · 2026-05-27

**Summary Of Contributions:**

The paper proposes a benchmarking framework to evaluate the shape-based recognition capability of object recognition models., the benchmark includes ~70k grayscale images of 200 objects rendered from multiple viewpoints. The underlying idea is that probing the embedding space of multiple viewpoints, and observing their clustering assignment via nearest-neighbor matching, can indicate whether the model attends to the shape itself, or other non-shape cues such as textures. The analysis spans across 321 pre-trained models with diverse architectures and reveals that many models are biased by non-shape cues, indicating better training techniques should be developed to remove these biases.

**Additional Comments:**

**Strengths**:
* Sound motivation.
* Extensive experiments with a lot of pre-trained models (supervised and self-supervised) with rigorous analysis on ResNet50.
* Interesting insights about the decision boundaries in modern object recognition models.


**Weaknesses**:
* The underlying assumption that shape matters most in recognition systems is not entirely valid when, for example, the color/texture does affect the output (e.g., detecting venomous snakes).
* No scale variations, i.e., objects do not change their size in the dataset (I think that even small changes would benefit, as large changes will indeed affect the low-level details).


**Minor**:
* Figures 7-11: please mention in the caption that these are results for the ResNet mentioned in Section 3.1
* Page 15: “table 1” -> “Table 1”

**Audience:**

Yes

**Audience Explanation:**

Yes, the object recognition/detection community, and computer vision practitioners in general, would be interested in the findings of this paper.

**Claims And Evidence:**

Yes

**Claims Explanation:**

Yes, the claims are supported by clear evidence. See below for strengths.

**Requested Changes:**

* Question: How is the benchmark going to be distributed? Open-source on GitHub?
* Please address “Weaknesses” and “Minor”.

---

> ### Author Response · Authors · 2026-07-04
>
> # Overall comment
> **The paper proposes a benchmarking framework to evaluate the shape-based recognition capability of object recognition models., the benchmark includes ~70k grayscale images of 200 objects rendered from multiple viewpoints. The underlying idea is that probing the embedding space of multiple viewpoints, and observing their clustering assignment via nearest-neighbor matching, can indicate whether the model attends to the shape itself, or other non-shape cues such as textures. The analysis spans across 321 pre-trained models with diverse architectures and reveals that many models are biased by non-shape cues, indicating better training techniques should be developed to remove these biases.**
>
> We thank the reviewer for these comments. We would choose slightly different wording in describing the role that non-shape cues play in ShapeY’s measurement scheme.  The main non-shape variable we have explored is 3D viewpoint; any OR system worth its salt must perceive the similarity of different 3D views of the same object, at least over a certain range. The effect of this variable is quantified in the ShapeY benchmark by measuring error rates as a function of the viewpoint exclusion radius.  ShapeY can also quantify the effects of what we call “appearance” changes, such as changes in object color, texture, or lighting. The effects of these kinds of changes are measured through the use of appearance exclusions.  This is illustrated in Figure 6 and 9, where the non-shape change was a reversal of background shade from black to white.
>
> **Question: How is the benchmark going to be distributed? Open-source on GitHub?**
> We plan to publicly release ShapeY on GitHub upon acceptance. Because the generated image set (68,200 images) is too large to host directly on GitHub, it will instead be distributed via a Google Drive link referenced from the repository's README. For the purposes of review, we have already made an anonymized version of the repository available:
> https://anonymous.4open.science/r/ShapeYV2-548F/README.md

---

> ### Author Response · Authors · 2026-07-04
>
> ## Comment 1
> **The underlying assumption that shape matters most in recognition systems is not entirely valid when, for example, the color/texture does affect the output (e.g., detecting venomous snakes).**
> We agree that texture and color provide critical information for classification, and that a system relying on shape alone would fail to distinguish a venomous snake by its markings. However, in human vision this kind of surface cue operates at the subordinate level of categorization (e.g., distinguishing a venomous coral snake from a harmless king snake), while shape remains the foundational cue at the basic level (e.g., recognizing something as a snake at all). Our work aims to quantify how well a vision system's representations align with this basic-level, shape-based organization, which is why we focus on shape rather than the full set of cues used across all levels of categorization. We refer to this basic-level/subordinate-level distinction explicitly in the third paragraph on page 2, where we note that the venomous-snake example falls under subordinate-level classification, rather than basic-level shape-based recognition that is ShapeY’s focus.
>
> ## Comment 2
> **No scale variations, i.e., objects do not change their size in the dataset (I think that even small changes would benefit, as large changes will indeed affect the low-level details).**
> We thank the reviewer for this suggestion. Scale is indeed a natural transformation to add to future versions of ShapeY, and we have noted it explicitly as a planned extension (see below). For the current version of the benchmark, scale changes were not included because with the small size of the images, reducing the scale of the objects leads to a loss of detail (similar to blurring an image), which can therefore be considered as a type of image degradation.  Given that the effects of image degradations on recognition performance have already been studied extensively in datasets such as ImageNet-C, we did not prioritize this type of transformation in our first release of the benchmark.
> As mentioned above, we plan to expand ShapeY's set of image transformations in the next iteration to include scale changes, as well as non-rigid transformations such as changes to an animal’s body configuration. We have updated the paper's final paragraph to explicitly state this.
>
> ## Comment 3
> **Figures 7-11: please mention in the caption that these are results for the ResNet mentioned in Section 3.1; Page 15: “table 1” -> “Table 1”**
> We have updated the captions of Figures 7–11 to note that all results are for the pre-trained ResNet50 introduced in Section 3.1, and corrected the capitalization of "Table 1" on page 15.

---

### Review · Reviewer_CYeC · 2026-06-20

**Summary Of Contributions:**

The contribution of this work is twofold: first, the compilation of a dataset of shape images corresponding to different views of 3D objects; and then, the evaluation of several OR models for tasks of retrieval and classification of the compiled dataset. Authors also claim that they provide a suite of quantitative and graphical tools for analyzing the structure of the embedding space of this dataset, but such tools are not really presented in the paper.

**Additional Comments:**

- The abstract and introduction section present a clear context and a convincing problem identification.
- The list of properties for the dataset is adequate.
- The definition of PCM's and the exclusion zone seems adequate.
- Although the results are expected, this type of study is well needed and much appreciated. Analyses like this one about dataset similarity must be conducted more often on academic datasets and for sure must be performed in real-world applications.

**Audience:**

Yes

**Audience Explanation:**

The overall idea is of high relevance to the community at large. The document will be much stronger if the experimental protocol is presented with a more formal structure and justification.

**Broader Impact Concerns:**

No comments on this regard.

**Claims And Evidence:**

No

**Claims Explanation:**

Claims about the need of the dataset, and the characteristics of the compiled and proposed benchmark are correct. However, the claims about the graphical tools for analyzing the embedding space are not presented.

Moreover, the claims and findings from the experiments need stronger justification. Concretely, the definition of the different experiments must be improved with a better structure, and the goals of each experiment must be presented more clearly.

**Requested Changes:**

- Provide a url to the suite of graphical tools for the evaluation of the embedding space.
- Although the approach of evaluating the performance at the level of object and category is adequate, a stronger justification and definition of how the categories are formed must be provided.
- In Figure 7, it is unclear the difference between No and 0 exclusion radio. Please provide a clarification.
- It is unclear why authors chose ResNet18 as the default model when there are lighter CNN models, or more recent Transformer models that have proven to obtain better performance on mainstream tasks. This is somehow confirmed in Figure 17.
- It is unclear why authors decided to change from ResNet18 to ResNet50. Please provide a justification.
- Section 3.6 mentions that the embedding space requires fixing. They probably meant that it requires a projection to a separable space.
- Regarding the results presented in Table 2. Please clarify whether this experiment maintains the same exclusion criteria as the previous one? Does the improvement depend merely on the training of the linear layer or is it due to a modification of the definition of the dataset? Please, also explain what is the train-val-test split?
- The experiment of section 3.7 must be clarified. How can we have ResNet trained using self supervised models?
- In section 3.8, for the escenario c), does it correspond to Contrastive Learning? If so, please use the standard name.
- Also, for the training of the experiment in section 3.8, please indicate how the values of the several hyperparameters were stablished? Was there any validation procedure applied? And what are the non-linear activation functions of those models?
-


Minor requests:
- Define all acronyms at their first occurrence, including DN.
- Figure 5 is mentioned before Figure 4 in the main text. Also Figure 8 before Figure 7. Whenever possible, mention images in a sequential order to ease reading. Additionally, place all objects (images, tables, equations, etc) right after the paragraph when they are mentioned for the first time.

---

> ### Author Response · Authors · 2026-07-04
>
> We would like to thank the reviewer for providing a thorough review and detailed comments.
>
> ## Comment 1
> **Claims about the graphical tools for analyzing the embedding space are not presented.**
> ShapeY's various displays do speak to the properties and quality of an OR system's embedding space; this reasoning appears in the manuscript in the "Geometric perspective of ShapeY performance measures" and "Qualitative assessment of ShapeY matching errors" sections. For example, examining the matching grids produced during a ShapeY run reveals several things about the embedding space:
> * When Top-1 error rates are high, this means the embedding space is of generally low quality for shape-based matching.
> * When category error rate is low but object error rate is high, the embedding space is well structured for shape at the coarse scale, but not at the fine scale.
> * When both category and object error rates are low, the identity of the best-matching view becomes informative: if the best match is consistently the view at the closest eligible viewpoint relative to the reference image (given the current viewpoint exclusion), as opposed to an unpredictably varying viewpoint, this means that distance in the embedding space is systematically related to the physical and perceptual difference between the two views. This desirable condition manifests as well-organized columns in the matching grids.
> * Matching grids also support judgments of error severity: some matching errors are "forgivable," in that the erroneous match happens to be highly visually similar to the reference image.  On the other hand, visually egregious errors indicate "tangles" in the embedding space (DiCarlo & Cox, 2007), which are highly undesirable.
>
> ## Comment 2
> **Provide a url to the suite of graphical tools for the evaluation of the embedding space.**
> https://anonymous.4open.science/r/ShapeYV2-548F/README.md
> The full suite of quantitative and graphical tools (error-rate graphs, viewpoint tuning curves, positive/negative match-score histograms, and ordered best-match grids) is shown in Figures 7–12 and released, with documentation, in the anonymized repository linked above. Upon acceptance, we will update the link to our actual GitHub repository.
>
> ## Comment 3
> **Although the approach of evaluating the performance at the level of object and category is adequate, a stronger justification and definition of how the categories are formed must be provided.**
> The 200 objects fall into 20 basic-level categories (airplane, chair, faucet, and so on), each with 10 object instances. These categories are drawn from ShapeNet's model collection (Chang et al., 2015), which, like ImageNet, organizes its 3D models according to WordNet's hierarchy of noun synsets. Because WordNet synsets span every level of generality—from superordinate concepts (e.g., "furniture") to specific subordinate ones (e.g., "rocking chair")— the specific synsets used in ShapeY were selected to correspond to basic-level concepts in the human conceptual hierarchy. Basic-level categories make sense as the grouping here because they're defined by rough shape similarity among their members (property 6 in Section 2.1), which is exactly the level at which human shape-based recognition operates.
>
> ## Comment 4
> **In Figure 7, it is unclear the difference between No and 0 exclusion radius. Please provide a clarification.**
> "No exclusion" means “No viewpoint exclusion”, so that every 3D view of the object counts as an eligible positive match including the reference view itself. This case makes sense only when an appearance exclusion is also applied.
> A viewpoint exclusion of 0 means that only the reference view is excluded as a positive match candidate, leaving any and all transformed views of the object eligible for matching. We have added this explanation explicitly to the manuscript (page 7-8), both in the main text where the exclusion radius is first defined and in the caption of Figure 7 itself.

---

> ### Author Response · Authors · 2026-07-04
>
> ## Comment 5
> **It is unclear why authors chose ResNet18 as the default model when there are lighter CNN models, or more recent Transformer models that have proven to obtain better performance on mainstream tasks. This is somehow confirmed in Figure 17. It is unclear why authors decided to change from ResNet18 to ResNet50. Please provide a justification.**
> There seems to be a slight mix-up here: all of our primary experiments were run on ResNet50, and ResNet18 does not appear anywhere in the manuscript, so there isn't a ResNet18-to-ResNet50 transition to explain. To rule out any inconsistency on our end, we've re-checked the manuscript to confirm ResNet50 is referenced consistently throughout, and have also updated the captions of Figures 7-12 to state explicitly that these results are for ResNet50.
> We chose ResNet50 to introduce and demonstrate ShapeY's analysis tools because it provides a widely familiar baseline.  A broader comparison across 321 pre-trained architectures, including lighter CNNs, ConvNeXt, and modern transformers (ViT, CaiT, XCiT, and DINOv2), appears in Section 3.9 and Figure 16, where DINOv2 achieves the best performance at 62% object-level accuracy. Notably, Figure 16 also shows that ResNet50, and convolutional architectures more generally, are not inherently inferior to transformers on ShapeY: plain ViTs actually underperform ResNet50, and only some transformer variants surpass it, generally due to specific training objectives (e.g., distillation) or, in DINOv2's case, a substantially larger pretraining dataset, rather than the transformer architecture itself. This suggests that ShapeY performance depends on the interaction of architecture, training objective, scale, and pretraining dataset, rather than on architecture family alone.
>
> ## Comment 6
> **Section 3.6 mentions that the embedding space requires fixing. They probably meant that it requires a projection to a separable space. Regarding the results presented in Table 2. Please clarify whether this experiment maintains the same exclusion criteria as the previous one? Does the improvement depend merely on the training of the linear layer or is it due to a modification of the definition of the dataset? Please, also explain what is the train-val-test split?**
> We apologize for the ambiguity. By "the embedding requires fixing," we meant that we hold the pre-trained embedding fixed (frozen) and train a single linear readout on top of it—i.e., a linear probe projecting the frozen 2048-dimensional embedding to a linearly separable class space. We have changed the wording to "freezing" to elminate this ambiguity, and changed the section header to "ResNet50's nearest-neighbor matching failures do not stem from an out-of-distribution problem" to state our finding more clearly.
> Regarding Table 2: linear probing is a classification task, distinct from our main nearest-neighbor matching task, so the viewpoint exclusion-radius criterion (which is specific to matching) does not apply here. The reported improvement comes solely from training the linear layer.  The frozen embedding and the ShapeY image set are otherwise unchanged, so the result reflects class information already linearly present in the pre-trained embedding, rather than any modification of the dataset.
> There is no train/val/test split for this experiment: since our goal was only to test whether class information is linearly decodable from the frozen embedding, rather than to assess generalization to held-out objects, we trained and evaluated the linear probe on the full ShapeY image set.
>
> ## Comment 7
> **The experiment of section 3.7 must be clarified. How can we have ResNet trained using self supervised models?**
> Happy to clarify: these are ResNet50 backbones, the same architecture as our supervised model, whose weights come from training with self-supervised objectives (SimCLR, MoCo, BYOL, SwAV, VICReg, Barlow Twins, DINO, and others) rather than from supervised cross-entropy training on ImageNet labels. Only the training objective changes across these models; the underlying architecture is ResNet50 in every case. We obtained these pre-trained weights from the LightlySSL library. We have also made this distinction explicit in Section 3.7 of the manuscript, which now states that the ResNet50 backbone is architecturally identical to our supervised baseline, but pre-trained with a self-supervised objective instead of supervised cross-entropy on ImageNet labels.

---

> ### Author Response · Authors · 2026-07-04
>
> ## Comment 8
> **In section 3.8, for the scenario c), does it correspond to Contrastive Learning? If so, please use the standard name.**
> Scenario (c) isn't contrastive learning in the InfoNCE sense, since it uses no negative pairs. It's SimSiam (Chen & He, 2021), which maximizes the cosine similarity between two augmented views of the same object using a stop-gradient and a predictor MLP. The manuscript already refers to this method by its standard name, "SimSiam," throughout Section 3.8, including in the description of the fine-tuning scenarios and the accompanying results discussion. Contrastive learning in the sense the reviewer may have in mind is instead discussed more generally in Section 3.7, in relation to self-supervised pre-training methods.
>
> ## Comment 9
> **Also, for the training of the experiment in section 3.8, please indicate how the values of the several hyperparameters were established? Was there any validation procedure applied? And what are the non-linear activation functions of those models?**
> For cases (a) and (b), we fine-tune with SGD (learning rate 5e-4, momentum 0.9) for 20 epochs. For SimSiam (c), we use SGD (learning rate 0.0025, scaled to batch size 64), momentum 0.9, and weight decay 5e-4, with a two-layer projection MLP (2048→1024→2048). These are standard recipes for ResNet50 fine-tuning and for SimSiam, not the result of a per-experiment hyperparameter search; our generalization check instead comes from the held-out objects (i) and held-out categories (ii) splits, run 5 times each with reported 95% confidence intervals. The non-linear activations are ReLU throughout, in both the ResNet50 backbone and the SimSiam projection MLP.
>
> ## Comment 10
> **Define all acronyms at their first occurrence, including DN.**
> We've now defined all acronyms at first use, including DN (deep network), OR (object recognition), VT (viewpoint transformation), PMC (positive match candidate), and OCD (obviously completely different).
>
> ## Comment 11
> **Figure 5 is mentioned before Figure 4 in the main text. Also Figure 8 before Figure 7. Whenever possible, mention images in a sequential order to ease reading. Additionally, place all objects (images, tables, equations, etc) right after the paragraph when they are mentioned for the first time.**
> We've reordered the text and figure placement so figures are referenced in sequence (Figure 4 before 5, Figure 7 before 8, and so on), and each figure, table, and equation now appears right after the paragraph that first mentions it.

---

> > ### Comment · Reviewer_CYeC · 2026-07-14
> > **Reviewing**
> >
> > A big thank you to the authors for their kind replies to my observations, and for the effort improving their manuscript. All answers about comments 4 to 6 and 8 to 11 are pertinent and clear. However, there are still a few things to be clarified.
> >
> > - Regarding comments 1 and 2: It seems that authors use the term "Graphical tools for analyzing the embedding space" when they meant simply Figures, as opposed to other more sophisticated tools like interactive dashboards. Please be clear in the document, changing the term tools for Figures. Moreover, those Figures are the standard way to show evidence in research papers, and therefore, not a concrete contribution.
> >
> > - For comment 3: I understand the provided explanation, and it seems correct. Please indicate where in the manuscript those explanations have been incorporated.
> >
> > - For comment 7: If I understand correctly, you took the ResNet50 architecture, randomly initialized, and trained it using the loss functions of the different self-suppervised approaches. Is that right? If so, please specify any modification to the architecture, the loss functions, and the treatment applied to the input data.